# Tools for BIM-GIS Integration (IFC Georeferencing and Conversions): Results from the GeoBIM Benchmark 2019

**Francesca Noardo** [1,*] , **Lars Harrie** [2] , **Ken Arroyo Ohori** [1] , **Filip Biljecki** [3,4] , **Claire Ellul** [5] , **Thomas Krijnen** [1] , **Helen Eriksson** [2] , **Dogus Guler** [6] , **Dean Hintz** [7] , **Mojgan A. Jadidi** [8] , **Maria Pla** [9] , **Santi Sanchez** [9] , **Ville-Pekka Soini** [2] , **Rudi Stouffs** [3] , **Jernej Tekavec** [10] and **Jantien Stoter** [1]

1    3D Geoinformation, Delft University of Technology, 2628BL Delft, The Netherlands; k.ohori@tudelft.nl (K.A.O.); t.f.krijnen@tudelft.nl (T.K.); j.e.stoter@tudelft.nl (J.S.)
2    Department of Physical Geography and Ecosystem Science, Lund University, S-223 62 Lund, Sweden; lars.harrie@nateko.lu.se (L.H.); helen.eriksson@nateko.lu.se (H.E.); villepekka.soini@gmail.com (V.-P.S.)
3    Department of Architecture, National University of Singapore, Singapore 117566, Singapore; filip@nus.edu.sg (F.B.); stouffs@nus.edu.sg (R.S.)
4    Department of Real Estate, National University of Singapore, Singapore 117566, Singapore
5    Department of Civil, Environmental and Geomatic Engineering, University College London, London WC1E 6BT, UK; c.ellul@ucl.ac.uk
6    Department of Geomatics Engineering, Istanbul Technical University, 34469 Istanbul, Turkey; gulerdo@itu.edu.tr
7    Safe Software, Surrey, BC V3T 0M1, Canada; dean.hintz@safe.com
8    Geomatics Engineering, Lassonde School of Engineering, York University, Toronto, ON M3J 1P3, Canada; mjadidi@yorku.ca
9    Institut Cartogràfic i Geològic de Catalunya, 08038 Barcelona, Spain; maria.pla@icgc.cat (M.P.); santi.sanchez@icgc.cat (S.S.)
10   Faculty of Civil and Geodetic Engineering, University of Ljubljana, 1000 Ljubljana, Slovenia; jernej.tekavec@fgg.uni-lj.si
*    Correspondence: f.noardo@tudelft.nl

**Abstract:** The integration of 3D city models with Building Information Models (BIM), coined as GeoBIM, facilitates improved data support to several applications, e.g., 3D map updates, building permits issuing, detailed city analysis, infrastructure design, context-based building design, to name a few. To solve the integration, several issues need to be tackled and solved, i.e., harmonization of features, interoperability, format conversions, integration of procedures. The GeoBIM benchmark 2019, funded by ISPRS and EuroSDR, evaluated the state of implementation of tools addressing some of those issues. In particular, in the part of the benchmark described in this paper, the application of georeferencing to Industry Foundation Classes (IFC) models and making consistent conversions between 3D city models and BIM are investigated, considering the OGC CityGML and buildingSMART IFC as reference standards. In the benchmark, sample datasets in the two reference standards were provided. External volunteers were asked to describe and test georeferencing procedures for IFC models and conversion tools between CityGML and IFC. From the analysis of the delivered answers and processed datasets, it was possible to notice that while there are tools and procedures available to support georeferencing and data conversion, comprehensive definition of the requirements, clear rules to perform such two tasks, as well as solid technological solutions implementing them, are still lacking in functionalities. Those specific issues can be a sensible starting point for planning the next GeoBIM integration agendas.

**Keywords:** georeferencing; conversions; interoperability; CityGML; Industry Foundation Classes; Building Information Models; 3D city models; standards

## 1. Introduction

In recent years, the integration of 3D geoinformation (3D city models) and Building Information Models (BIM), coined as GeoBIM, has become an important topic, addressed by a growing community coming from several fields within academy (geoinformation, geomatics, construction, architecture and urban planning) as well as from organizations outside the academy (government-related institutions, National Mapping and Cadastral Agencies, private companies, etc.).

The exchange of information between geospatial (3D city models) and BIM sources enables the reciprocal enrichment of the two kinds of information with advantages for both fields, e.g., automatic updates of 3D city models with high-level-of-detail features, automatic representation of BIM in their context, automated tests of the design, and so on [1–12].

The GeoBIM and, more generally, the integration challenge is composed of several issues:

1. The harmonization and consistency of data themselves are the first requirement, which have to concretely fit together, with similar or harmonizable features (e.g., accuracy, geometric and semantic representation, amount of detail, georeferencing).
2. Interoperability is fundamental to develop a reliable and reproducible integration methodology. The metadata must be clear and comprehensive, and data formats have to be understood and correctly interpreted uniquely by both humans and any supporting software. Moreover, an interoperable dataset is supposed to remain unchanged when going through several imports and exports by software tools, possibly converting it to their specific native formats and exporting it back. In this regards, to facilitate common understanding and agreed rules, it is desirable to use open standards.
3. Effective conversion among different formats has to be allowed, i.e., transforming one dataset in a likely standardized format to another one in full compliance with the end format specifications and features. Within this point, both previous concepts of harmonization of typical features of the resulting representation and interoperability of the produced format (validity of geometry, consistency of semantics and so on) have to be taken into account.
4. Finally, the data providers and procedures need to change to use the integrated GeoBIM information instead of BIM and GIS separately. Collaborations with many stakeholders and actors are generally necessary for this step.

In order to evaluate how effective current software are to solve the issues of interoperability and conversion, the GeoBIM benchmark project (https://3d.bk.tudelft.nl/projects/geobim-benchmark/) was proposed and launched in 2019, funded by the International Society for Photogrammetry and Remote Sensing (ISPRS) and the European Association for Spatial Data Research (EuroSDR). In particular, the aim was to get a better picture of the state of software support for the two most used open standards for 3D city models and BIM, i.e., respectively CityGML by the Open Geospatial Consortium (OGC) [13], and the Industry Foundation Classes (IFC) by the buildingSMART consortium [14]. According to the experience of the authors, and exchanged in several informal occasions with a wider community, there are some shortcomings in the use of these two standard data formats and especially how they are handled by software. The aim of the benchmark was to investigate the software interoperability for CityGML and IFC [15]. This was studied via two tasks: Task 1: What is the support for IFC within BIM—and other—software? and Task 3: What is the support for CityGML within GIS—and other—tools?. The results of these two tasks will be investigated in detail in a different paper.

The first aim of this paper is to study georeferencing, which was Task 2 in the benchmark. The georeferencing of BIM models is a central issue in GeoBIM that is related to both harmonization (quality of referencing information) and interoperability (how this information is interpreted and used by software). This study focuses on the harmonization side of the problem, namely how software can

read and write georeferencing information in IFC files. This issue is especially important since IFC supports several ways to store georeferencing information (Section 2.2.3).

The second aim of this paper is to evaluate tools for conversion (Task 4). Again, it is a matter of both harmonization (transformation of model features) and interoperability (output of valid files that can be read in other software). This is discussed in Section 2.3. The evaluation is important since there is an increasing number of tools developed to convert CityGML to IFC and especially IFC to CityGML. Important issues to evaluate here are the conversion capability of the tools and how the end result of the conversion varies between them.

The paper is organised as follows. Section 2 is devoted to describe the three first sub-issues for integration: harmonization (Section 2.1), interoperability (Section 2.2) and conversions (Section 2.3), with some available solutions and potential shortcomings and gaps. Then, in Section 3, the set-up of the benchmark and methodology of the evaluation is described. The result of the georeferencing interoperability is reported in Section 4, and the result of the conversion tools study is found in Section 5. The paper ends with concluding remarks and reminders for future works (Section 6).

## 2. Integration Background

### 2.1. Harmonization: 3D City Models vs BIM Features

Within the field of GeoBIM, it is apparent that 3D city models and BIM differ in several aspects (see Table 1), e.g., De Laat and Van Berlo [16]. This implies that to integrate the two models for any application (besides pure visualization), it is important that their features are consistent and aligned. Such harmonization would entail that all of the data correspond to the requirements of one of the two models, either 3D city models or BIM. They could also be something else, if a third-party use case would be identified. In this last case, both models should change to meet the new requirements properly.

**Table 1.** Comparison between 3D city models features and Building Information Model (BIM) aspects.

|  | 3D City Models | BIM |
|---|---|---|
| *Geometry* | mainly boundary representation (explicit) | mainly parametrically modelled solids (implicit) |
| *Main data source* | survey of real world objects | design |
| *Approximate range of detail (d)* | $1000 > d > 0.1$ m | $50 > d > 0.001$ m |
| *Semantics* | aimed at the description of city/landscape representation | aimed at the description of small building elements representation |
| *Georeferencing* | compulsory | optional |
| *Supported analysis and decisions* | city-level | building-level |
| *Evolution of* | Geographical Information Systems (GIS) | Computer-Aided Design (CAD) |
| *Dominated by* | government | industry |

One of the most apparent inconsistencies between the two models refers to georeferencing. While georeferencing is long-standing practice in GIS in general and in 3D city models in particular, it is a fairly new feature for BIM. Designers usually work in a local Cartesian system, without a more complex management of global/spherical coordinates system being required. 3D modelling, computer-aided design (CAD) and BIM software can give issues when working far from the coordinates' origin, as it happens in georeferenced systems. For this reason, the storage of georeferencing information was not a priority within BIM. Therefore, more powerful entities to manage it in IFC are added only in the most recent versions (Section 2.2.3).

## 2.2. Interoperability and Standard Data Formats

Interoperability is fundamental for the accurate interpretation and use of data within systems and tools as well as for the re-use and exchange of data. Hence, integration is strongly supported by the transparency and explication of the data and metadata; this is where standards, and especially open standards, have their vital role. Several standards and open data models are produced to represent, exchange and support integration in the 3D city models field and in the BIM field. Some international examples are the data models proposed by the European Directive for an Infrastructure for Spatial Information in Europe (INSPIRE); gbXML; OGC LandInfra. Among these, the most acknowledged and used open standards are CityGML by OGC (Section 2.2.1), for 3D city models, and IFC by buildingSMART for BIMs (Section 2.2.2). On a national level, there are specific models which often are based on, or at least linked to, the international standards [17,18]. Therefore, these open standards are being considered as reference guideline for investigating the GeoBIM integration in most studies (e.g., Sun et al. [11], Daum et al. [19]) and are chosen in a joint effort between OGC and buildingSMART to integrate geodata and BIM data.

### 2.2.1. The OGC CityGML

CityGML (by OGC) [20] is the most internationally widespread standard to store and exchange 3D city models with semantics in the geospatial domain. It structures the description of the geometry and semantics of city objects.

In the most recent developments within the CityGML working group, the data model and its implementation are considered separately, according to the suggestions coming from the developers community as explained in the following text. For this reason, also in this description the two parts are treated separately.

CityGML is traditionally implemented as an application schema for the Geography Markup Language (GML). CityGML uses version 3.1.1 of GML [21]. It is an open format and it is human readable. That means that the information could potentially be retrieved even if losing backwards compatibility in software. However, GML presents issues from a software developer point of view (e.g., regarding the geometry representation, see http://erouault.blogspot.com/2014/04/gml-madness. html). The consequences of this were pointed out by Task 3 of the benchmark too [15].

To overcome the issues, alternative implementations were proposed, such as a Structured Query Language (SQL) database—PostgreSQL, in 3DCityDB [22], and more recently in JavaScript Object Notation (JSON), within CityJSON 1.0 (https://www.cityjson.org) [23], based on the CityGML 2.0 data model. These options are intended to improve usability and effectiveness of the CityGML data model.

The GML implementation is considered the official one, recommended by the standard. Consequently, the studies and tools supporting the conversions were also keeping it as reference. For this reason, it is the one used for conversions within this benchmark study as well.

CityGML 2.0 (current version) contains classes structured into 12 modules, each of them extending the core module, containing the most general classes in the data model, with city object-specific classifications, e.g., Building, Bridge, WaterBody, CityFurniture, LandUse, Relief, Transportation, Tunnel, Vegetation. The most developed and most used module in practice is the Building module, which is also the one where 3D city models and BIM foremost meet, although recent extension of the BIM scope are more and more including the infrastructure field as well.

The semantic data model of CityGML is being updated, with the proposed version 3.0 (https://github.com/opengeospatial/CityGML-3.0CM). Some features in the version 3.0 are intended to make the CityGML model closer to BIM. The main change with respect to version 2.0 is the addition of a new space concept [24]. The new space concept can be utilized to define, using inheritance, new classes e.g., "*BuildingConstructiveElement*". This class offers possibility to store detailed building elements, typical of BIM. A similar approach to opening elements (e.g., solids voiding walls) and filling elements (e.g., windows, doors) is applied in the 3.0 data model as well. How to populate the

new classes, and how to convert/align the geo-concepts in the CityGML data model into the new BIM-oriented classes is the next question to solve interoperability.

CityGML geometries are essentially the same for all classes: objects are represented as boundary surfaces embedded in 3D and consist of triangular and polygonal faces. No change of geometry management is proposed for version 3.0.

2.2.2. Industry Foundation Classes (IFC)

The buildingSMART Industry Foundation Classes (IFC) standard (https://technical. buildingsmart.org/standards/ifc/) is an open standard data model for BIM to be shared and exchanged through software applications, domains and use cases, within the Architecture, Engineering and Construction (AEC) and Facility Management (FM) fields. It includes classes for describing both physical and abstract concepts (e.g., cost, schedule, etc.) concerning AEC-FM for mainly buildings. Planned new versions extend it to include infrastructures and other kinds of constructions (https://technical.buildingsmart.org/standards/ifc/ifc-schema-specifications/). It has also been adapted as the ISO 16739 international standard [25]. The standard includes relevant constructs for a wide variety of disciplines, use cases and processes associated to the construction domain, most prominently the semantic description and geometric representation of typical construction elements and their relationships.

IFC is structured in a hierarchical data model, furthermore organized in several, deep and complex meronymic (part-of) trees too. The spatial composition (Site/Building/Storey/Space/Zone) is one more kind of aggregation, different from the element (part-of) composition one (e.g., a stair and the assembled elements in it). Moreover, nesting is a slightly different kind of element composition, representing the products which are specifically designed as complementary by manufacturers. Finally, subtraction relationships are also part of the IFC model, representing openings by means of the voiding mechanism. A great number of further relationships are added to this complexity (e.g., to associate materials, geometric representation or other property information and so on).

An additional complexity to the semantic model is given by the possibility to store the same kind of object by means of several entities. For example, the layers within a compound wall object can be represented by means of an associated IfcMaterialLayerSet, but also as a more generic decomposition where every wall layer is modelled as a distinct IfcBuildingElementPart. Furthermore, a great number of attributes can be associated with entities, and inherited from the parent-ones, both directly or through property sets.

All this semantic complexity is intended to represent faithfully the buildings as functional to the standard designed scope. However, the implementation and use of such a theoretically precise model is difficult and can result in inaccuracies or under-use of it, besides hindering interoperability for leaving too much freedom in filling the information in and in choosing the kind of representation to be used.

Additional terms, which can be used in IFC, are defined within the buildingSMART Data Dictionaries (bSDD) and are modelled according to the International Framework for Dictionaries (IFD) (http://bsdd.buildingsmart.org). It is based on the standard ISO 12006-3.

The IFC current versions are: IFC2x3, which was released in 2005 (with the latest corrigendum in 2007) and the IFC4.1 from 2018. At the time of writing, the most implemented and used version is still IFC2x3 by far. For this reason, both versions were considered in this benchmark study.

IFC derives many aspects from ISO 10303 [26], informally known as STEP. The majority of geometry definitions are derived from ISO 10303-42 and the typical exchange formats are based on STEP Physical File (SPF, ISO 10303-21) and an XML encoding (ISO 10303-28).

Parametric modelling is usually employed in BIM and IFC, which makes it possible to encode many kinds of geometries. This includes Boolean operations and complex sweeps, for example the sweep of an arbitrary profile along a curve while constraining the normal vector. In addition, explicit geometries are supported in the form of Boundary Representations and (added in IFC4)

efficient support for triangulated meshes. The implementation of the former type of geometry is that supporting the full stack of geometric procedures in IFC is a major implementation effort and due to implementation choices can sometimes lead to different results in importing applications. The complexity can, therefore, have consequences on interoperability and the way different pieces of software read and re-export the same geometry.

This high level of complexity could be challenging to ensure consistency in the use of the model, including conversion to and from other formats.

### 2.2.3. Georeferencing IFC Files

Proper georeferencing of an IFC file allows the link between the model of a single building or construction within its context and environment. There are several options to store georeferencing information in IFC, with varying level of detail as described by Clemen and Hendrik [27]. These options range from basic address information to the definition of an offset between the project coordinate system and the global origin of a coordinate reference system (CRS) and the corresponding rotation of the XY-Plane (Table 2). By considering models coming from practice, it is possible to notice that the *LoGeoRef20* option is the most commonly used within exported IFC models by BIM software. *LoGeoRef30* is not an officially defined georeferencing method in the IFC standard but is sometimes adopted by software tools and practitioners. *LoGeoRef50* is the best georeferencing options from a geodata/geomatics perspective, but is still quite uncommon in practice for building IFC models (likely more commonly used in infrastructure BIM models). It is usually complicated to understand if and how an IFC file is georeferenced, although some tools (e.g., the *IfcGeoRefChecker* at https://github.com/dd-bim/IfcGeoRef) are available to check this.

**Table 2.** List of georeferencing options in Industry Foundation Classes (IFC) classified as *LoGeoRefs* [27].

| LoGeoRef | Supported CRS | Storing Entities |
|---|---|---|
| *LoGeoRef10* | No CRS, approximate location by means of the address. | *IfcPostalAddress* referenced by either *IfcSite* or *IfcBuilding*. |
| *LoGeoRef20* | WGS84 EPSG:4326 | Attributes *RefLatitude*, *RefLongitude*, *RefElevation* within *IfcSite* |
| *LoGeoRef30* | Any Cartesian CRS, including projected coordinates (CRS not specified in the file) | *IfcCartesianPoint* referenced within *IfcSite* (defining the projected coordinates of the model reference point); *IfcDirection* attribute of *IfcSite* (stores rotations regarding project or global north. (Ad-hoc solution used by several tools.) |
| *LoGeoRef40* | Any Cartesian CRS, including projected coordinates (CRS not specified in the file) | Attribute *WorldCoordinateSystem* storing the coordinates of the reference point in any Cartesian CRS (including the projected ones) and direction *TrueNorth*. Both are stored within *IfcGeometricRepresentationContext*. (Most official IFC2x3-way to store the reference system.) |
| *LoGeoRef50* | Specific projected CRS, specified by means of the EPSG code | IFC v.4 only. Coordinates of the reference point stored in *IfcMapConversion* using the attributes *Eastings*, *Northings* and *OrthogonalHeight* for global elevation. Rotation for the XY-plane stored using the attributes *XAxisAbscissa* and *XAxisOrdinate*. The CRS used is specified by *IfcProjectedCRS* in the attribute *Name* by means of the proper EPSG code. |

The georeferencing of BIM has not been a priority for architects and software developers. Therefore, the topic of georeferencing and CRSs, traditionally belonging to the field of geomatics and cartography, has only recently reached the architectural representation world and the BIM tools. Another difficulty for the BIM tool developers is the diversity of georeferencing options used in IFC (as

noted in Table 2). As a consequence, architects and modellers do not regularly store and use accurate georeferencing information.

In the benchmark, the ability of tools for IFC georeferencing was investigated from two perspectives (see Section 3.1):

1. interpretation of the georeferencing information provided in the IFC file,
2. editing capabilities of the georeferencing information.

The focus of the benchmark was therefore to study the methods to import and create georeferencing information. Another perspective that was not studied is the capability of the tools to utilize this information to optimize the georeferencing. This is a non-trivial issue especially since the coordinate system in the IFC model is based on a local Cartesian system while the CRS is based on an ellipsoidal Earth model (which will cause the scale to vary within the model). Detailed geodetic aspects of georeferencing is e.g., explored by Uggla and Horemuz [28] and a pragmatic method for retrieving georeferencing information for visualisation is found in Diakité and Zlatanova [29].

*2.3. Conversions*

Conversions from BIM to 3D city models and from 3D city models to BIM should both deal with the interoperability and harmonization issues. Therefore the resulting model should be both geometrically and semantically valid, with respect to the chosen output format. In addition, the features of the resulting model must be consistent with the ones foreseen for the output format (see Section 2.1). An optimal conversion procedure would allow the selection of specific characteristics, according to the model within which such result is supposed to be integrated. For example, if a BIM is supposed to be integrated in an LoD1 city model (according to the CityGML LoDs), the result of conversion should be a boundary representation of the generalization of the building as a footprint extruded to a height. Although more challenging, even the other way around should be true: the boundary representation of a 3D city model, if converted to BIM should achieve thick solid walls and modelled details, as far as possible and sensible for the specific use case considered.

In addition to the off-the-shelf software able to make conversions, several studies were developed to propose conversion methodologies. Most of them focus on IFC to CityGML conversions e.g., Arroyo Ohori et al. [7], Stouffs et al. [9], Donkers et al. [30], Olsson [31], Yu and Teo [32]. The ones here cited are the attempts, among a huge literature (e.g., Liu et al. [1]), considering the harmonization of respective features, including both semantics and geometry, and sometimes digging into the generalization of different levels of detail of CityGML.

However, as mentioned before, by Arroyo Ohori et al. [7], a thorough methodology converting both semantics and geometry consistently with the 3D city model features, as useful for the use of the model for analysis, had little success in previous efforts. In addition, it is quite common to find issues in the resulting data quality, consisting of semantics and geometric inaccuracies, inconsistency or loss of information, use of the wrong spatio-semantic paradigm in the resulting models, besides possibly more serious errors of invalidity, misplacement or deformations [33].

Fewer studies investigated the conversions from a CityGML model to IFC (e.g., Salheb [34]).

The tests carried out within the GeoBIM benchmark aimed at the assessment of the quality of the models resulting from current conversion tools, in both directions, from IFC to CityGML and from CityGML to IFC, in terms of both validity of the produced models (interoperability) and transformation and mapping of element features, consistently with the resulting output (harmonization).

## 3. Methodology

*3.1. The GeoBIM Benchmark General Set-Up*

Processing abilities are often developed in software in full compliance with the reference data models, which would guarantee them to work with the compliant models, in an ideal world. However,

the results can be different when using the tools for current models developed within practice. For this reason, four topics were defined, to be further investigated in the GeoBIM benchmark project, using datasets modelled in practice, to assess their actual effectiveness:

**Task 1** What is the support for IFC within BIM (and other) software?
**Task 2** *What options for geo-referencing BIM data are available? (This Task is the first objective of this paper.)*
**Task 3** What is the support for CityGML within GIS (and other) tools?
**Task 4** *What options for conversion (software and procedural) (both IFC to CityGML and CityGML to IFC) are available? (This Task is the second objective of this paper.)*

To facilitate these tasks a set of representative IFC and CityGML datasets were provided [35] and used by external, voluntary, participants in the software or procedure they would like to test [6]. Full details about the tested software and a full list of participants can be found in the respective pages of the benchmark website (https://3d.bk.tudelft.nl/projects/geobim-benchmark/software.html for the tested software and https://3d.bk.tudelft.nl/projects/geobim-benchmark/participants.html for the list of participants.).

There were no expertise nor skill requirements to participate in the benchmark tests. Therefore, some information could be wrong or inaccurate, due to little experience with the tested software or the managed topics. The declared level of expertise is intended to lower this possible bias. Moreover, the benchmark team and the authors tried to double check the answers (at least the most unexpected ones) as much as possible, but the answers reported in the data were generally not changed from the original ones.

*3.2. The Provided Datasets*

A number of datasets from several organisations were identified, pre-processed and validated for this benchmark activity (Table 3—see Noardo et al. [35] for details). The datasets were chosen to test both the most common features of such data and the main detected issues regarding the interesting but tricky aspects of the format.

**Table 3.** Provided CityGML and IFC data for the GeoBIM benchmark 2019.

| Name | Description | Dimension | Source | Aim |
|---|---|---|---|---|
| CityGML v.2 Amsterdam .gml | Seamless city model covering the whole city of Amsterdam, including several CityGML city entities (vegetation, roads, water, buildings, and so on). Level of Detail (LoD) 1. | 4.06 GB | Generated through 3dfier by TUDelft (https://github.com/tudelft3d/3dfier) | Test of the hardware-and-software connected performances (it is a very heavy model), and support for the included city classes. |
| CityGML v.2 Rotterdam -LoD12.gml | Textured CityGML model of one district in Rotterdam, including only Buildings in LoDs 1 and 2. | 33.91 MB/154.4 MB (with textures) | Municipality of Rotterdam (NL) | Test of the support for multiple LoDs and textured files. |
| CityGML v.2 Buildings -LoD3.gml | Procedurally modelled buildings in LoD 3 | 1.33 MB | Generated through Random3Dcity (https://github.com/tudelft3d/Random3Dcity) [36]. | Test of the support for LoD 3 files and related classes. |
| IFC v.2x3 Myran.ifc | Model of a small 2-floor building in Sweden, by Swedish architects. Georeferenced. | 27.14 MB | MONDO arkitekter, Falun, (SE) | Test of the main functionalities of software and common procedures. |

| Name | Description | Dimension | Source | Aim |
|---|---|---|---|---|
| IFC v.2x3 UpTown.ifc | Model of a big complex tower in Rotterdam, by Dutch architects. | 241.04 MB | Municipality of Rotterdam (NL) | Test of the software's performance. |
| IFC v.4 Savigliano.ifc | Model of a building in Italy, by an Italian architect within a research environment. | 21.55 MB | Arch. Lorenzo Polia (IT) | Test of the support for IFC v.4 and to enable the tests of procedures and tools working with IFC v.4 |
| IFC v.2x3 and 4 Specific IFC geometries | Set of geometries modelled using a range of the modelling alternatives allowed in IFC, which are often not supported or incorrectly interpreted by software. | 0.31 MB | Generated on purpose with IfcOpenShell | Test of the support and behaviour of software with respect to these specific geometries. |

### 3.2.1. IFC Geometry Sets

The geometries used in the BIM models can have a huge numbers of variations and fully checking them and their consistency and correctness is still an unsolved task. Moreover, IFC allows a number of geometry types that can be useful to modellers, but they are sometimes not supported and can be interpreted in different ways by software. On the other hand, IFC puts validity constraints on certain geometries. Some software has implemented workarounds to read those invalid geometries too, which are often undocumented. Consequently, there is often little possibility to keep track of these solutions. For these reasons, a specific set of geometries (Table 4, Figure 1) was provided among the benchmark datasets in order to test the specific cases.

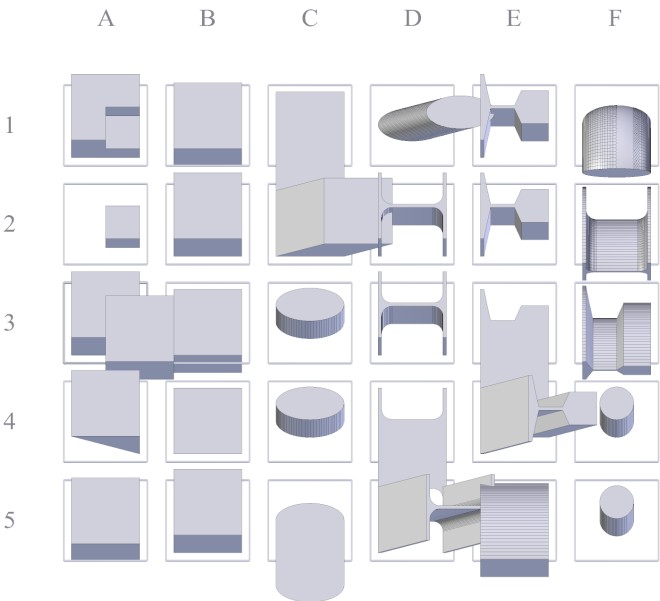

**Figure 1.** Reference schema of the modeled IFC geometries.

**Table 4.** Description of each object included in the IFC geometries set, in both IFC2x3 and IFC 4. The geometries written in italic are not included in the IFC 4 dataset since dismissed.

| | IFC Definition | Description | Valid |
|---|---|---|---|
| A1 | IfcBooleanResult_1 | Result of Boolean subtraction with two cube operands with partial overlap | Yes |
| A2 | IfcBooleanResult_2 | Result of Boolean intersection with two cube operands with partial overlap | Yes |
| A3 | IfcBooleanResult_3 | Result of Boolean union with two cube operands with partial overlap | Yes |
| A4 | IfcBooleanClipping Result_1 | Result of Boolean clipping operation with a cube and a halfspace operand | Yes |
| A5 | IfcShellBasedSurface Model_1 | A shell based surface model, an explicit collection of faces | Yes |
| B1 | IfcFacetedBrep_1 | A faceted boundary representation, an explicit collection of faces | Yes |
| B2 | IfcExtrudedAreaSolid_1 | Extrusion of a rectangular profile | Yes, Rectangle Profile in CV2.0, normalized depth |
| B3 | IfcExtrudedAreaSolid_2 | Extrusion of a rectangular profile, negative extrusion depth | No, violation of where rule Depth >0 (- 2) |
| B4 | IfcExtrudedAreaSolid_3 | Extrusion of a rectangular profile, zero extrusion depth | No, violation of where rule Depth >0 (0) |
| B5 | IfcExtrudedAreaSolid_4 | Extrusion of a rectangular profile, non-normalized direction vector | Yes, Rectangle Profile in CV2.0, non-normalized depth |
| C1 | IfcExtrudedAreaSolid_7 | Extrusion of a rectangular profile, direction vector parallel to profile | No, violation of where rule dot product <>0 |
| C2 | IfcExtrudedAreaSolid_10 | Extrusion of a rectangular profile, slanted direction vector | Yes |
| C3 | IfcExtrudedAreaSolid_13 | Extrusion of an elliptical profile | Yes, Ellipse Profile in CV2.0, normalized depth |
| C4 | IfcExtrudedAreaSolid_16 | Extrusion of an elliptical profile, non-normalized direction vector | Yes, Ellipse Profile in CV2.0, non-normalized depth |
| C5 | IfcExtrudedAreaSolid_19 | Extrusion of an elliptical profile, direction vector parallel to profile | No, violation of where rule dot product <>0 |
| D1 | IfcExtrudedAreaSolid_22 | Extrusion of an elliptical profile, slanted direction vector | Yes |
| D2 | IfcExtrudedAreaSolid_25 | Extrusion of an I-shape profile | Yes, I-shape profile in CV2.0, should have fillets, normalized depth |
| D3 | IfcExtrudedAreaSolid_28 | Extrusion of an I-shape profile, non-normalized direction vector | Yes, I-shape profile in CV2.0, should have fillets, non-normalized depth |
| D4 | IfcExtrudedAreaSolid_31 | Extrusion of an I-shape profile, direction vector parallel to profile | No, violation of where rule dot product <>0 |
| D5 | IfcExtrudedAreaSolid_34 | Extrusion of an I-shape profile, slanted direction vector | Yes, should have fillets, |
| *E1 (Not in the IFC 4 dataset)* | IfcExtrudedAreaSolid_37 | Extrusion of a crane rail (A-shape) profile | Not required Crane Rail Profile not in CV2.0 (Not in IFC4) |

**Table 4.** *Cont.*

|  | IFC Definition | Description | Valid |
|---|---|---|---|
| *E2* (Not in the IFC 4 dataset) | IfcExtrudedAreaSolid_40 | Extrusion of a crane rail (A-shape) profile, non-normalized direction vector | Not required Crane Rail Profile not in CV2.0 (Not in IFC4) |
| *E3* (Not in the IFC 4 dataset) | IfcExtrudedAreaSolid_43 | Extrusion of a crane rail (A-shape) profile, direction vector parallel to profile | No, violation of where rule dot product <>0 (Not in IFC4) |
| *E4* (Not in the IFC 4 dataset) | IfcExtrudedAreaSolid_46 | Extrusion of a crane rail (A-shape) profile, slanted direction vector | Not required Crane Rail Profile not in CV2.0 (Not in IFC4) |
| E5 | IfcRevolvedAreaSolid_1 | Revolution of a rectangular profile | Yes, Revolved Solid in CV2.0 |
| F1 | IfcRevolvedAreaSolid_2 | Revolution of an elliptical profile | Yes, Revolved Solid in CV2.0 |
| F2 | IfcRevolvedAreaSolid_3 | Revolution of an I-shape profile | Yes, Revolved Solid in CV2.0, should have fillets (toroidial surfaces in this case) |
| *F3* (Not in the IFC4 dataset) | IfcRevolvedAreaSolid_4 | Revolution of a crane rail (A-shape) profile | Not required Crane Rail Profile not in CV2.0 (Not in IFC4) |
| F4 | IfcSweptDiskSolid_1 | Swept disk | Yes, Swept Disk Solid in CV2.0 |
| F5 | IfcSweptDiskSolid_2 | Swept disk with parameter range outside of curve definition | No Parameter range outside of curve definition |

*3.3. Answer Templates for the Participants*

The participants contributed in two ways to the tasks evaluated in this paper, i.e., Task 2 and 4. Firstly, they filled an online form designed as answer template, where they described the actions taken in the tool and the required setting and parameters (Sections 3.3.1 and 3.3.2). The aim of this was primarily to produce a guide to people intending to use the same tool for the intended scope (i.e., georeferencing or data conversion). In addition, participants could add comments and observation to discuss both their results, specifically and possibly related issues to the more general topic of each task.

The second contribution from the participants was the obtained result files themselves, allowing the benchmark team to inspect and analyse them to point out the relevant observations.

3.3.1. Answer Template for Task 2: Georeferencing

In Task 2, the query form was divided into two parts—more details are found at https://3d.bk. tudelft.nl/projects/geobim-benchmark/task2.html. In the first part, the participants were asked to measure the processing times and smoothness of user interactions, namely, viewing actions (panning and zooming) as well as simple analyses/queries to investigate if this time was dependent on whether the BIM data were referenced or not. The rationale behind this was that there is an advantage of working with a local coordinate system since the coordinate values will be smaller, which might affect the software's smoothness depending on how the implementation is done. For instance, the use of large coordinates with floating-point arithmetic can cause flickering issues due to the loss of precision in transformations.

The second part of the query form concerned the description of functionality of the georeferencing, such as whether the georeferencing is a built-in functionality or if a plugin is required. Other questions concerned how the CRS were handled. Can the program handle all CRS? Are the CRS on project or object level? It should be noted that Task 2 does not include questions about whether and how georeferencing information can be imported/exported from IFC/CityGML files, since these issues were addressed by Task 1.

### 3.3.2. Answer Template for Task 4: Conversions

The Task 4 template was simple, since a great variety of tools with completely different procedures could be used. It was required to give a description of the tool used, adding details about the settings and possibly necessary precautions to obtain good results, together with an approximate assessment of the processing times needed. If a bespoke piece of software was used, instead of an off-the-shelf tool, documentation and references about it had to be added. In addition, the form included some place for open discussion, both as space for free comments and by asking explicitly if any change in the original files could have made the conversion easier. This part of the form—details are found at https://3d.bk.tudelft.nl/projects/geobim-benchmark/task4.html—has a value especially as documentation and reference for whom intends to use similar tools.

Since the tools can change significantly from one to the other, it is only possible to draw few direct conclusions from this. However, more interesting results can be outlined by the analysis of the delivered converted models (Section 3.4.2), which are directly comparable, even considering that there is no reference solution of a 'perfect' conversion.

### 3.4. Assessment Methods for the Delivered Models

### 3.4.1. IFC Georeferencing (Task 2)

In the first part of the assessment the information filled in the answer template was evaluated and summarised when possible. The main interest in this phase lies in the description and the efficiency of the tools.

In the second part of the assessment, the resulting georeferencing of the models delivered by the participants was checked and referred to the right level of georeferencing (LoGeoRef), see Section 2.2.3.

We did not check the consistency between the georeferencing information possibly stored multiple times, such as both in the *IfcSite* attributes according to *LoGeoRef 20* and also as foreseen by *LoGeoRef 30*. We suppose that the different alternatives could validly coexist and be used by different software according to the degree of accuracy required by the use case (e.g., approximation for energy analysis or GeoBIM integration for planning and city design support).

From this part of the results it is possible to see how the possibilities allowed by the IFC standard are at present implemented in tools.

### 3.4.2. IFC to CityGML and CityGML to IFC Conversions (Task 4)

The most relevant part in the materials delivered by participants for Task 4 are the converted models.

Their analysis entailed several challenges. One main challenge is the lack of ground truth that could be regarded as reference for the evaluation. Since there is no conversion methodology that can be regarded as perfect, no such ground truth exists. To overcome this issue, an analysis was planned, based on comparison between the models, where the reciprocal discrepancies were used, and the possible ground truth was defined by starting from the obtained statistical parameters. However, the converted models are still far from having a sufficient quality to be compared to some very refined ground truth. Therefore, they were inspected and analysed by means of 3D viewers in order to check the appearance of geometry and semantics and to manually check consistency and correctness with

respect to the destination standard. Furthermore, the delivered converted models were validated with respect to geometry and semantics, when tools were available.

In the case of resulting CityGML data, the delivered models were validated by means of the geovalidation tools available at http://geovalidation.bk.tudelft.nl: val3dity (which applies tests to the geometry) and the CityGML schema checker. Moreover, their visualization within two 3D viewers was checked: azul (https://github.com/tudelft3d/azul) [37] and the FZK Viewer (https://www.iai.kit.edu/english/1648.php). The products of the conversion by means of the *IFC2CityGML* tool, producing CityGML v.3 data could not be checked, since not supported by the used software; only the validation of the geometry could be verified.

For the systematic validation of IFC schema and geometry, no known comprehensive tool is available. Therefore, we also had to rely on the manual inspection of the generated models within several 3D viewers, in order to reduce the bias given by possible inaccuracies in the interpretation of the model by the software itself. Solibri Model Viewer (https://www.solibri.com), RDF IFCViewer (http://rdf.bg/product-list/ifc-engine/ifc-viewer/), FZK Viewer (https://www.iai.kit.edu/english/1648.php), BIMVision (https://bimvision.eu/en/download/).

## 4. Results of Georeferencing (Task 2)

### 4.1. Tested Software to Georeference IFC Models

Table 5 lists the six software packages used in the test.

**Table 5.** Software tested for IFC georeferencing (Task 2).

|  | Open Source | Proprietary | Freeware |
|---|---|---|---|
| **GIS Software** [1] |  | ESRI ArcGIS Pro [2] |  |
| **'Extended' 3D viewers** [3] |  | CSTB eveBIM [4] | FZK Viewer [5] |
| **ETL and conversion software** [6] | IfcGeoRefChecker | FME Desktop [7] |  |
| **BIM software** [8] |  | Autodesk Revit v.2020 [9] |  |

[1] GIS combine different kind of geodata and layers and make analysis on them, structured in a database, in a holistic system. [2] https://www.esri.com/en-us/arcgis/products/arcgis-pro/overview. [3] Software originally developed for visualising the 3D semantic models, including georeferencing, and query them. They were (sometimes later) extended with new functions for applying symbology or making simple analysis. [4] https://www.evebim.fr. [5] https://www.iai.kit.edu/1648.php. [6] Extract, Transform and Load (ETL) software, and conversion software, are able to apply transformations or computations to data. [7] https://www.safe.com. [8] Intended to design buildings or infrastructures according to the the Building Information Modelling methods. [9] https://www.autodesk.com/products/revit/overview?plc=RVT&term=1-YEAR&support=ADVANCED&quantity=1.

*IfcGeoRefChecker* (https://github.com/dd-bim/IfcGeoRef/) is a dedicated tool developed by the Dresden University of Applied Sciences for georeferencing IFC files. Two tests were performed using *Autodesk Revit* (2019 and 2020 versions) and two tests using *FME Desktop*: one of the *FME* tests was made by an expert at Safe Software (from here on "FME_Desktop") and one test was based on an *FME* script developed for georeferencing of IFC files (from here on "FME_script").

### 4.2. Tools for Georeferencing IFC: Notes from the Participants Reports

The questions in the first part considered the time performance for visualization and queries/analyses, and especially if there were any performance differences between a non-georeferenced and a georeferenced model. The results (Find the complete delivered forms at https://www.dropbox.com/sh/3ezbcjm8zlysz95/AADqkTeRQGy8tPawBrgcNaLpa) show that for all the software, the performance is not dependent on whether the BIM model was georeferenced or not, since the tools probably store the georeferencing information as a value, but follow working in the same local system, without translating and rotating the whole system to the georeferenced point, and even less projecting the model in a cartographic system.

All software used georeferencing tools without using any plugin or similar.

In some cases, it is possible to set the required CRS and even a height reference system. However, the destination IFC file, especially in version 2x3, is not always able to store such metadata. Some more advanced packages, such as *FME*, foresee the possibility of making more sophisticated conversions between different CRS, as usual for geoinformation related tools. Additionally, in this case, the issue is the final storage of the result within the suitable entities and attributes in the IFC model.

The georeferencing worked in different ways. For example, *eveBIM* was able to import the georeferencing from IFC files by using either the *IfcLocalPlacement* (default setting) or the multi scale setting (using the *IfcSite* georeferencing, including the *TrueNorth*, *IfcLocalPlacement* and the *IfcSite* elevation, or a personalized option).

Several programs allow interactive georeferencing. For example in *Revit* the user can specify the geodetic coordinates of the origin of the local Cartesian system. Furthermore, most of the software allow the user to rotate the model as part of the georeferencing (towards the CRS north), exception here were e.g., *eveBIM* and *IfcGeorefChecker*. Most pieces of software allow several CRS to be defined, by e.g., using EPSG codes, although the list of supported CRS can vary. The limitation is mainly on the height component. For example, there is no support of the height reference system in *FZK Viewer*.

### 4.3. The Exported Georeferenced IFC Models: Which LoGeoRef?

The models exported by four applications were delivered (Find them at https://www.dropbox.com/sh/rqvk7x2w0zacxsr/AADd9jqyhM1ymhHUE0UIizPxa) (Tables 6 and 7): *eveBIM*, *FZK Viewer*, *Revit* and *FME*. *IfcGeoRefChecker* was reported to be able to export the georeferenced model as well, but for some reason the model was not delivered. It could also be the only tool considering the LoGeoRef as defined by Clemen and Hendrik [27], since the authors are the same.

**Table 6.** Results of the applied georeferencing procedures to the models, exported in IFC v.2x3 by participants.

| IFC 2X3 Models | LoGeo Ref North East | LoGeo Ref Elev. | Rotation | Notes |
|---|---|---|---|---|
| *eveBIM–Myran* | 30 | 30 | 40: *TrueNorth* stored (−0.534, 0.845) | |
| *eveBIM–UpTown* | 30 | 30 | 0: Not stored, *TrueNorth* set as (0, 1) | Instead of referencing an additional Cartesian point containing the georeferenced coordinates, it changes the origin coordinates usually stored at object #6 in the IFC STEP file. |
| *FME_Desktop–Myran* | 20 | 0 | 0: Not stored/default *TrueNorth* value | However, the projected coordinates are used to translate the reference points of several objects and are stored accordingly into the file. WGS84 coordinates are stored in the *RefLatitude* and *RefLongitude*. |
| *FME_script–Myran* | 30 | 30 | 40: *TrueNorth* stored(−0.534352349, 0.845261833) | |
| *FME_script–UpTown* | 30 | 30 | 30: Direction stored with reference to the *IfcSite* −0.28652455272779837, 0.9580728994623191, 0. | Elevation is stored with the x, y coordinates in the point referenced to *IfcSite*. The wrong value, 450 instead of −1.5, is probably due to human inaccuracy, being the same value than for the Savigliano model |

**Table 6.** *Cont.*

| IFC 2X3 Models | LoGeo Ref North East | LoGeo Ref Elev. | Rotation | Notes |
|---|---|---|---|---|
| *Revit–Myran* | 0 | 0 | 0: Not stored/default TrueNorth value | Coordinates stored as referenced to *IfcSite*, but their values are wrong: 33869.3163757324, 41055.4707641601, −1200.00000915529 |
| *Revit–UpTown* | 30 | 20 | 30: Direction stored with reference to the IfcSite −0.286524552727797, −0.95807289946232, 0 | |
| *Revit–Savigliano 2x3* | 0 | 0 | 0: Not stored / default TrueNorth value | Coordinates stored as referenced to *IfcSite*, but their values are wrong: 0, 0, 2.35 |
| *FZK–Myran* | 30 | 30 | 0: Directions are general 0, 0, 1 or 1, 0, 0 | The coordinates are reported several times (11 in total), associated to *IfcSite*, *IfcBuildingStoreys* and other *IfcElements*. |
| *FZK–UpTown* | 30 | 30 | 0: Directions are general 0, 0, 1 or 1, 0, 0 | The coordinates are reported several times (500 in total), associated to *IfcSite*, *IfcBuildingStoreys* and other *IfcElements*. |
| *FZK–Savigliano 2x3* | 30 | 30 | 0: Directions are general 0, 0, 1 or 1, 0, 0 | The coordinates are reported several times (12 in total), associated to *IfcSite*, *IfcBuildingStoreys* and other *IfcElements*. |

**Table 7.** Results of the applied georeferencing procedures to the models, exported in IFC v.4 by participants.

| IFC 4 Models | LoGeo Ref North East | LoGeo Ref Elev. | Rotation | Notes |
|---|---|---|---|---|
| *FME_script–Savigliano* | 30 | 30 | 30+40: TrueNorth stored (−0.814115518, −0.580702956); Direction referred to IfcSite stored (0.14176513680170655, 0.9899003212382514, 0) | No *IfcMapConversion* nor *IfcProjectedCRS* is there. |
| *Revit–Savigliano* | 0 | 0 | 0 | No *IfcMapConversion* nor *IfcProjectedCRS* is there. |
| *FZK–Myran4* | 30 | 30 | 0: Directions are general 0, 0, 1 or 1, 0, 0 | Same as in the 2x3 version. The IFC4 *IfcMapConversion* reference to a Cartesian point 0, 0, 0. *IfcProjectedCRS* has value 'Unknown SRS'. It was set like this because the EPSG:3013, foreseen for the data, is not available in the software. |
| *FZK–UpTown4* | 30 | 30 | 0: Directions are general 0, 0, 1 or 1, 0, 0 | Same as in the 2x3 version. The IFC4 *IfcMapConversion* reference a Cartesian point 0, 0, 0. *IfcProjectedCRS* stores 'Local CRS', 'Local Cartesian Coordinate System', default value in *FZK Viewer*. |
| *FZK–Savigliano* | 30 | 30 | 0: Directions are general 0, 0, 1 or 1, 0, 0 | Same as in the 2x3 version. The IFC4 *IfcMapConversion* reference a Cartesian point 0, 0, 0). *IfcProjectedCRS* stores the correct CRS EPSG and name: 'EPSG:32632', 'WGS84 / UTM Zone 32N'. |

The highest *LoGeoRef* obtained is stored within the models exported by *eveBIM* and *FME_script*. The *LoGeoRef 30* criteria of storing projected coordinates within the cartesian point associated to *IfcSite* are followed for the 3 axis: North, East and Height. However, the storage of the correct direction varies in the different models. It is stored as *TrueNorth* attribute of the entity *IfcGeometricRepresentationContext* for the *Myran.ifc* model, by both tools. In the *Savigliano.ifc* model, only georeferenced by *FME_script*, both the *TrueNorth* attribute and the *IfcDirection* stored with reference to *IfcSite* are filled, but they have

different values. For the *UpTown.ifc* model, the direction is stored consistently with *LoGeoRef 30* in the *IfcDirection* associated to *IfcSite* by the *FME_script*.

Conversely, the *UpTown.ifc* model georeferenced by *eveBIM* presents an unusual choice: the point storing the georeferenced coordinates is associated to the *IfcSite*; however, such a point is not an additional one, but the *IfcCartesianPoint* is usually stored as object #6 in the IFC STEP file, which is representing the origin of the model. And the direction is not stored correctly (it has a value $(0, 1)$), making the quality of georeferencing a bit lower.

The models georeferenced by *FZK Viewer* come second, with *LoGeoRef 30*, but without storing the right direction: the present directions have general values $(0, 0, 1)$ or $(1, 0, 0)$. In addition, the projected coordinates are associated not only to the *IfcSite* entity, but to many others (e.g., IfcBuildingStoreys and other objects). That is also the reason why the files size increase so much: approximately three times in the cases of *Myran.ifc* and *Savigliano.ifc* and more than five times for the *Uptown.ifc* model.

With *Revit*, the georeferencing information was correctly stored only for the *UpTown.ifc* model. Specifically, North and East were added within the *IfcCartesianPoint* associated to *IfcSite* (as for *LoGeoRef 30*), whilst the Height was stored in the *RefElevation* attribute of *IfcSite* (as for *LoGeoRef 20*).

With the *Myran.ifc* and *Savigliano.ifc* models, the georeferencing through *Revit* instead failed.

*FME_Desktop* could associate a correct georeferencing, storing it as *LoGeoRef 20*: correct *RefLatitude* and *RefLongitude* attributes within *IfcSite* in WGS84. However, the *RefElevation* attribute has value 0. In addition, the projected coordinates are used to translate (added to) several *IfcCartesianPoints* associated to the objects in the file, inclusive of North East and Height.

Finally, the use of entities introduced by version 4 of IFC to allow a more accurate georeferencing, such as *IfcMapConversion* and *IfcProjectedCRS*, were checked for the models exported to IFC4 (Table 7). In the models exported by *FZK Viewer*, such entities are present in the file, but are not used consistently to apply the georeferencing: *IfcMapConversion* is just hosting default values, referencing a $(0, 0, 0)$ point in all the cases. Instead, the values of *IfcProjectedCRS* vary in the different files georeferenced by FZK Viewer: in *Myran.ifc* it is "Unknown SRS"; in *UpTowin.ifc* it is "Local CRS, Local Cartesian Coordinate system"; while in the *Savigliano.ifc* model the name and EPSG code of the projected CRS ('EPSG:32632', 'WGS84 / UTM Zone 32N') are filled correctly.

In the IFC4 *Savigliano.ifc* georeferenced model exported by *Revit* and by *FME_script* the entities added in the version 4 of the IFC data model to allow a more accurate georeferencing (*LoGeoRef 50*) are not used and not present in the exported file either.

## 4.4. Georeferencing IFC Discussion

This study tested some of the tools (raising in number) allowing the georeferencing of IFC files, which is being increasingly acknowledged as critical for the integration of the IFC data with 3D city models and other geospatial data, especially for the forthcoming applications related to infrastructure design. The relevance of this test lies also in the comparison of the currently implemented software and the level of research about the georeferencing in BIM (e.g., Clemen and Hendrik [27], Uggla and Horemuz [28]).

The results show how the storage of georeferencing parameters can follow different rules, as allowed by IFC, grouped by Clemen and Hendrik [27] in levels of georeferencing *LoGeoRefs* (see Table 2). However, the tools are generally not transparent on the way the georeferencing is applied, and therefore little control on this is possible by the user. The (LoGeoRefs) are not part of standard definitions supposed to be followed by software implementation. However, they could represent a consistent way of storing georeferencing parameters within IFC. Such storage rules are not always consistently followed among North-East coordinates, height values and rotation (different parameter can follow different *LoGeoRef*). In addition, in some cases, the same software stores georeferencing in different files according to different criteria.

With the tested tools, which are among the most advanced ones implementing the investigated processes, it is not possible to reach a *LoGeoRef* higher than 30 for the North-East and Height

values, although the *TrueNorth* attribute stores the rotation in some cases, according to the *LoGeoRef40* definition.

It can be discussed that different *LoGeoRefs*, especially *LoGeoRef 30* and *LoGeoRef 40*, present little differences in accuracy, but they mainly change for the foreseen storage of data. The preference for one of the two systems should be decided by the tools and applications using the models for specific processing. The importance of consistency of the stored information should be further studied based on such requirements, and it is not possible to judge it now.

Other two options, occurred among the results, could be considered with a similar approach starting from the use cases and processing software.

First option is the association of coordinates to many points within the file and not only to the entities considered by Clemen and Hendrik [27], or their addition to the model local ones. The possibilities enabled by this and how appropriate this could be also with respect to the computational consequences would need some study.

Similarly, the association of projected coordinates to the object #6 of IFC STEP files is not an official option, but a discussion about advantages/drawbacks of such a choice could be beneficial for future implementations.

Furthermore, we observe that few tools (or none) implement the theoretically more rigid option of using a combination of *IfcMapConversion* and *IfcProjectedCRS* allowed by IFC version 4. One reason is of course that this possibility is not included in IFC 2x3 which is still much used in practice.

In this study, only a limited amount of tools was tested and, unfortunately, few of them were able to export the obtained results. The overview that is offered is therefore limited, although the tested tools are probably the best ones for this specific task, many of them developed within geoinformation-experts groups. For example, few BIM softwares were tested, that usually offer some georeferencing tool, especially in the most recent releases. However, we could experience how the *LoGeoRef* they implement generally corresponds to the *LoGeoRef 20*.

In addition, this previously cited alignment between research and implementation of the georeferencing methods, in order to improve the possibility of importing and exporting georeferencing information, more recommendations to software vendors and users from official institutions, in particular buildingSMART, would be necessary. However, a first step towards interoperability and usability of models would be to agree with practitioners and developers about what the most effective *LoGeoRef* is for each processing or use case and related tools.

## 5. Results of Conversions (Task 4)

Generally there are few tools for conversions, and many of the tools described in the literature (cf. Section 2.3) are often not publicly available. If they are released as e.g., open source, they are most commonly not documented in such a way that they easily could be used by others. Therefore, few tools were used in this conversion task.

The results of conversions (Find the complete set of answers by participants at https://www.dropbox.com/sh/62yzpa5f3t3ty3f/AAASXHySqbtVFH-dOsY2i0cya?dl=0 and all the converted models by the tools at https://www.dropbox.com/sh/p0f7ds1dnbu0qsh/AACF06kT5yI2qEpuOSs52X2Ta?dl=0) from IFC to CityGML and from CityGML to IFC are analysed in the following Sections 5.1 and 5.2, respectively.

### 5.1. IFC to CityGML Conversions

For IFC to CityGML conversion, mainly three software were tested (Table 8) with different settings.

Most of the tests used *Safe Software FME* algorithms, directly or as plugins in third-party software, such as *ESRI ArcGIS*, with different scripts and configurations for the conversion (Some examples of how to use FME to make such conversion can be found at https://knowledge.safe.com/articles/591/bim-tutorial.html). Table 9 shows the details of such settings, together with the level of expertise of the participant using it (from L1—"beginner" to L4—"developer") (L1—Novice user (nearly the first

time using the software); L2—regular user; L3—expert user (knows very well technical details and less documented tricks); L4—developer of the tested software.). The column "Test ID" defines a code used in the following sections to report the related results. On the right part of the table, the converted datasets are reported.

**Table 8.** Software tested for conversions from IFC to CityGML (Task 4).

|  | Open Source | Proprietary | Freeware |
|---|---|---|---|
| **GIS Software** [1] |  | ESRI ArcGIS Pro [2] |  |
| **'Extended' 3D viewers** [3] |  |  |  |
| **ETL and conversion software** [5] | IFC2CityGML [4] | FME Desktop [6] |  |
| **BIM software** [7] |  |  |  |

[1] GIS combine different kind of geodata and layers and make analysis on them, structured in a database, in a holistic system. [2] https://www.esri.com/en-us/arcgis/products/arcgis-pro/overview. [3] Software originally developed for visualising the 3D semantic models, including georeferencing, and query them. They were (sometimes later) extended with new functions for applying symbology or making simple analysis. [4] [9]; https://ifc2citygml.github.io/. [5] Extract, Transform and Load (ETL) software, and conversion software, are able to apply transformations or computations to data. [6] https://www.safe.com. [7] Intended to design buildings or infrastructures according to the the Building Information Modelling methods.

**Table 9.** Software tested for conversions form CityGML to IFC (Task 4).

| Test ID | Software | Myran | UpTown | Savigliano | IfcGeom. | Ifc4Geom. |
|---|---|---|---|---|---|---|
| FME19-L3 | FME 2019 [L3] | 1 |  |  |  |  |
| FME19qt-L1 | Quick translator in FME 2019 [L1] | 1 | 1 | 1 | 1 | 1 |
| FME18qt-L1 | Quick translator in FME 2018.1 [L1] | 1 | 1 |  |  |  |
| FME18di-L1 | FME 2018.1 Data Inspector (Open dataset—save as CityGML file) | 1 |  |  | 1 |  |
| FME17-RVTr-L1 | FME 2017 with Revit reader [L1] | 1 |  |  | 1 | 1 |
| FME17-IFCr-L1 | FME 2017 with the IFC reader [L1] | 1 |  |  | 1 | 1 |
| AGIS-FMEqt-L2 | ArcGIS Pro [L2]—quick export tool |  |  | 1 |  |  |
| AGIS-FME-L1 | ArcGIS Pro—data interoperability extension adopting FME 2018.1 [L1] | 1 |  |  | 1 | 1 |
| AGIS-FME-IFCr-L1 | ArcGIS Pro—data interoperability extension adopting FME 2018.1—with IFC reader [L1] | 1 | 1 |  | 1 | 1 |
| AGIS-FME-RVTr-L1 | ArcGIS Pro—data interoperability extension adopting FME 2018.1—with Revit reader (which is denoted *IFC with Data Views (FME Exporter for Revit)* from *FME 2019* and onward; this reader can read both *Revit* and IFC files) [L1] | 1 | 1 |  | 1 | 1 |
| AGIS-FME-IFCdeprr-L1 | ArcGIS Pro—data interoperability extension adopting FME 2017 with the now deprecated version (from 2014) of the IFC reader [L1] | 1 |  |  | 1 | 1 |
| IFC2CityGML-L4 | IFC2CityGML [L4] |  |  | 1 |  | 1 |

Although the *FME Quick translator* is not recommended by the *Safe Software* vendors as a reliable option to perform complex transformations and conversions, it was used in several tests, mostly by non-expert participants. For this reason, the consequent results are included in this section as well. Some tests with *FME Quick translator*, in fact, reported that not all the features that were read were transformed (e.g., 3147 *IfcTypeObjects* out of 49826 features read for the *UpTown.ifc* dataset). Some explanation of this was given by the participants as for example, solids may be too complex, not closed or not orientable. Some solid geometry may be missing traits, appearance, measure or attributes. However, this has to be expected with such tool, mainly conceived to make simple format conversions (such as GIS-to-GIS), without the need of complex transformations or schemas mapping.

*ESRI ArcGIS Pro* was tested with its *Data interoperability extension* which implements an FME-based algorithm though, with the same readers as above.

Finally, the *IFC2CityGML* tool [9] was tested for IFC4 data (converting the data to CityGML v.3).

5.1.1. Specific IFC Geometries Conversion to CityGML

By inspection of the converted specific IFC geometry dataset, using 3D viewers, two main results were obtained:

**Case A**  Everything is *GenericCityObject* with *Lod4Geometry* (see Figure 2).
**Case B**  Everything is building with *lod4Solid* geometry, except for the yellow geometry (corresponding to the A5 geometry, see Section 3.2.1) which is an *lod4multisurface* geometry (see Figure 3)

Not all the objects in the IFCgeometries datasets were converted (see Figures 2 and 3). The geometries that were not converted are the ones generated by extrusions of crane rails (A-shapes), revolutions and swept disks. The A5 geometry, which is converted as *lod4multisurface* in the Case B, is the only geometry modelled as a shell (namely, shell-based surface model, an explicit collection of faces). The other elements are extrusions, revolutions or results of boolean operations between solids. One boundary representation is in the IFC dataset (B1), however it is not stored as a shell but as a faceted boundary representation.

For some reason, the datasets where the objects were interpreted as *bldg:buildings* with *lod4solid* geometry (Figure 3) appear to be mirrored with respect to the alternative conversion (as generic city object) as well as to the original file.

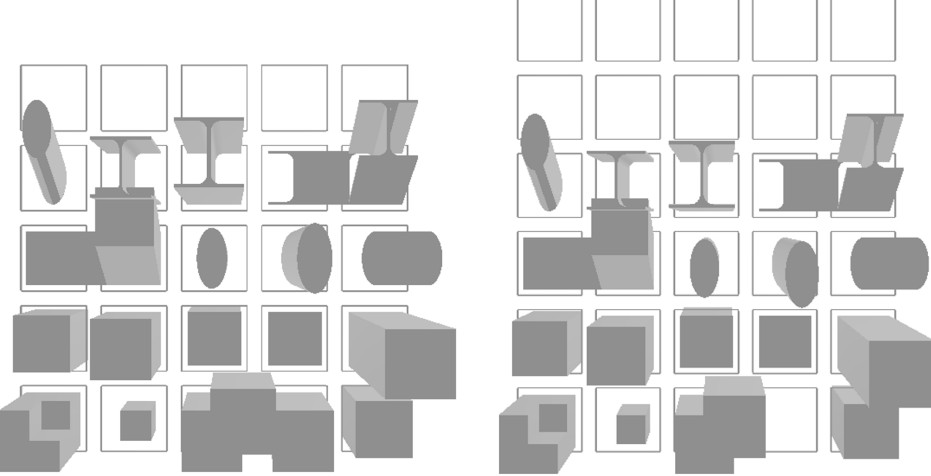

**Figure 2.** Case A: IFC geometries, both in IFC v.2x3 and v.4 converted to *Generic city objects*, visualized in azul.

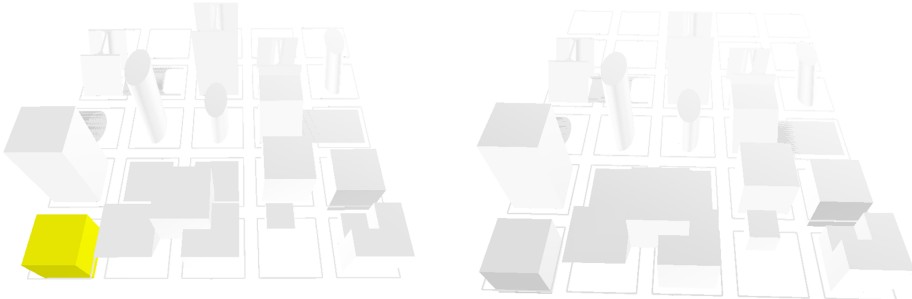

**Figure 3.** Case B: IFC geometries, both in IFC v.2x3 and v.4, converted to *bldg:buildings*, visualized in azul. It is possible to notice that, for some reason, these are inverted with respect to their original distribution.

Only this dataset could be analysed within the geovalidation tools, since the others were too heavy, converted from both the IFC 2x3 version and IFC 4 to CityGML v.2. In Tables 10 and 11 the most relevant results of the validations, and inspection, are summarised. It is possible to notice how few of the converted objects result in a valid geometry. Moreover, the results of *IFC2CityGML* tool, producing a CityGML v.3 file could not be tested by val3dity, since CityGML v.3 is not supported. The semantic schema is instead generally converted to a valid output (although it was very simple, for those datasets, including few simple entities).

**Table 10.** Analysis of the IFC4 geometries, in IFC v.4 dataset converted to CityGML. [1]

| Tested Software | Invalid 3D Primitives | Invalid Features | Schema Validity | Result Inspected Within azul and FZKViewer |
|---|---|---|---|---|
| *FME19qt-L1* | 32/45 [2] | 41/54 | Valid | Case A |
| *AGIS-FME-IFCdeprr-L1* | 29/45 | 29/45 | Valid | Case A |
| *AGIS-FME-IFCr-L1* | 43/44 | 43/44 | Valid | Case B |
| *AGIS-FME-RVTr-L1* | 43/44 | 43/44 | Valid | Case B |
| *FME17IFCr-L1* | 44/45 | 44/45 | Valid | Case B |
| *FME17RVTr-L1* | 44/45 | 44/45 | Valid | Case B |

[1] Find all the validation reports produced by val3dity at https://3d.bk.tudelft.nl/projects/geobim-benchmark/linkedfiles/T4/val3dityIFC4geom. [2] The ratio represents: number of valid objects / number of read objects.

**Table 11.** Analysis of the IFC geometries dataset, in IFC v.2x3 converted to CityGML. [1]

| Tested Software | Invalid 3D Primitives | Invalid Features | Schema Validity | Result Inspected within azul and FZKViewer |
|---|---|---|---|---|
| *FME18qt-L1* | 36/49 | 51/64 | Valid | Case A |
| *FME18di-L1* | 36/49 | 51/64 | Valid | Case A |
| *FME19qt-L1* | 37/50 | 51/64 | Valid | Case A |
| *FME17IFCr-L1* | 49/50 | 49/50 | Valid | Case B |
| *FME17RVTr-L1* | 49/50 | 49/50 | Valid | Case B |
| *AGIS-FME-IFCr-L1)* | 48/49 | 48/49 | Valid | Case B |
| *AGIS-FME-RVTr-L1)* | 48/49 | 48/49 | Valid | Case B |
| *AGIS-FME-RVTr-L1* | 33/49 | 33/49 | Non Valid: Element '{http://www.opengis.net/citygml/generics/2.0}value': is not a valid value of the atomic type 'xs:integer'., line 39 | Case B |

[1] Find all the validation reports produced by val3dity at https://3d.bk.tudelft.nl/projects/geobim-benchmark/linkedfiles/T4/val3dityIFCgeom.

### 5.1.2. UpTown.ifc Conversion to CityGML

The converted *UpTown.ifc* models (ranging between 1 GB and 1.48 GB) were too heavy to be validated with the previous tools. Therefore they were only inspected in the azul 3D viewer. In all the cases, everything is converted to *GenericCityObject* with *lod4geometry*. The information about the IFC entities is only stored in the attributes (e.g., Table 12). The attempt to visualize it in *FZK Viewer* failed and only a line could be visualized.

**Table 12.** Example of attributes associated to the *GenericCityObjects* in which the Uptown model was converted.

| Attribute | Value |
|---|---|
| GlobalId | 3188dEGGT9WfSyi$OfSoG2 |
| Name | Basic Wall: |
| ObjectType | Basic Wall: |
| Tag | 5490546 |
| ifc_parent_id | 0zrrkK2Jr1lv6ZAaGfUEZt |
| ifc_parent_uniqu... | 0zrrkK2Jr1lv6ZAaGfUEZt_259 |
| ifc_type_object_i... | 0jSzKjG_LEGwDaCh_otDw9_120132 |
| ifc_unique_id | 3188dEGGT9WfSyi$OfSoG2_966720 |

### 5.1.3. Savigliano.ifc Conversion to CityGML

The *Savigliano.ifc* model, in IFC 4, was successfully converted by *IFC2CityGML* [9] and the *FME Quick translator* (*FME19qt-L1*) as well. In addition, the dimension of the models (25.5 MB and 13.27 MB, respectively) does not allow to check their validity (geometric and semantic) by means of geovalidation tools.

When visualized in azul, we can see that the model converted by *FME* maintains all of its parts without any attempt of harmonization: no selection of objects nor change in representation (i.e., each parallelepiped wall is still represented by means of 6 connected surfaces even though it is no more a solid but a multi-surface). Instead, everything is converted to *GenericCityObject* with *lod4Geometry* as geometry, with similar pattern to the Case A described in Section 5.1.1. The top part of the building is missing (Figure 4).

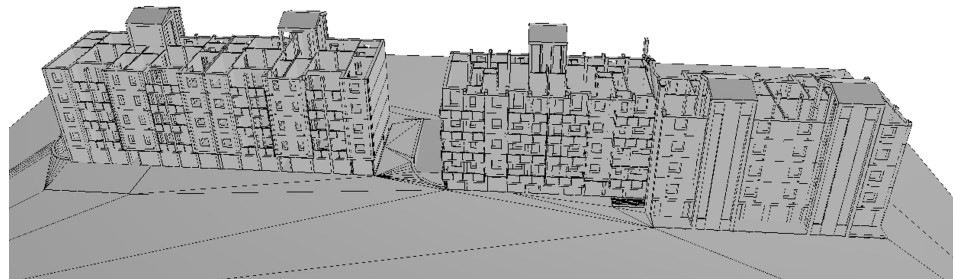

**Figure 4.** Example of Savigliano model converted to CityGML. Each object is converted to *GenericCityObject*, with *lod4Geometry*.

Similarly, the model converted by *IFC2CityGML* also loses its top part. Moreover, the participants doing the test also report the loss of some elements, after conversion with the rule set (Note that while *IFC2CityGML* supports multiple rule sets [38], only one rule set was available for this task.) (*IfcStairFlight*, *IfcSlab*, *IfcRailing*, *IfcDoor*, *IfcWindow*, *IfcBuildingelementProxy*). As they notice in addition, most missing elements seem to be *IfcClosedShell* objects, which had issues already when checking into *BIMserver*. On the other hand, the most successfully converted elements probably use

*IfcExtrudedAreaSolid* representation instead. One possible reason is the fact that the conversion was developed based on a limited set of models from practice and reflects on the elements available in these models.

In this case, the semantic mapping follows an approach that kept more semantics: the converted elements are *buildingConstructiveElements*, organized in *bldg:Storeys* and in one *bldg:Building* in turn, according to CityGML version 3 data model. The data model proposed by CityGML v.3 allows the validity of such kind of model, although it could be discussed that the spatio-semantic paradigm [33] is not changed in a harmonization effort with a GIS-consistent representation.

### 5.1.4. Myran Conversion to CityGML

All the converted Myran models were generally too large to be opened in the geovalidation tools (http://geovalidation.bk.tudelft.nl), or did not work properly anyway. However, in the cases where the semantic schema could be checked, it was valid.

When exported by *FME Quick translator*-based tools, the *Myran.ifc* model has a similar behavior than *Savigliano.ifc* and *UpTown.ifc*, maintaining most of the elements, without any selection (e.g., indoor-outdoor) and any kind of transformation in their representation. All the objects are converted to *GenericCityObject* with a generic *lod4Geometry*. Although it is not possible to understand what kind of geometry is used, apparently the solids are not converted to faces. However, as it was already discussed, the *Quick translator* is not the most proper *FME* tool to perform such a complex conversion, as recommended by the software vendors themselves.

With the *Myran.ifc* model, an attempt further was developed by two of the participants in order not to limit the conversion to the storage format but to actually map the semantics correctly between the two data models and change the kind of geometry used. In particular, this was done within the tests: FME19-L3; FME17-RVTr-L1; FME17-IFCr-L1; AGIS-FME-L1; AGIS-FME-IFCr-L1; AGIS-FME-RVTr-L1.

In the FME19-L3 test a CityGML LoD4 model was obtained (Figure 5) by means of a very complex workspace.

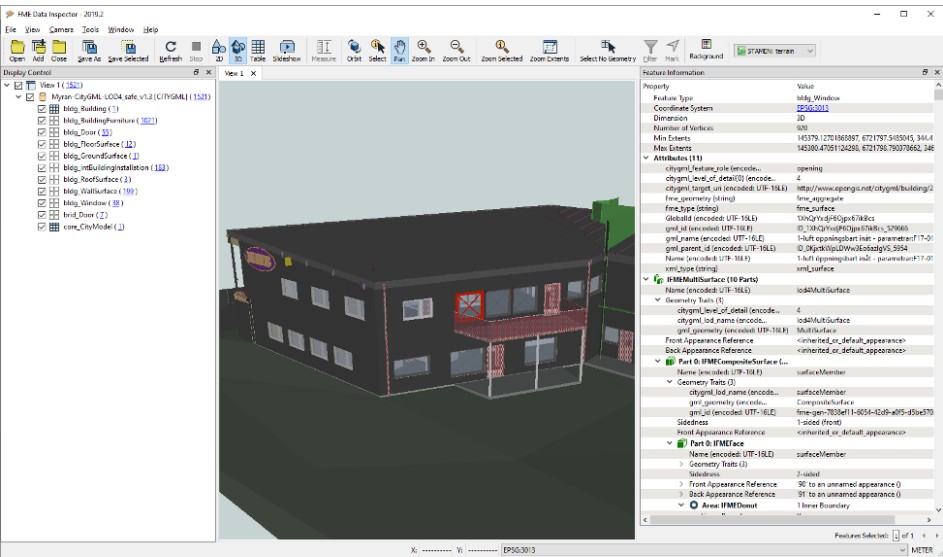

**Figure 5.** Views of the Myran model converted to CityGML by means of *FME 2019*.

In the other tests, the attempt is also done of getting to generalised boundary surfaces of the building for an LoD3 (According to the criteria followed by the CityGML standard)-like representation.

Among those last ones, the models that could be opened in viewers were quite similar. Doors are correctly converted to *bldg:doors*, windows are correctly converted to *bldg:windows*; roof is *roofSurface* and floor surface is converted to *bldg:floorSurface*. However, some of the elements (walls, stair, the railing

and the slab, the columns and pavement of the balcony and the signs) are (incorrectly) converted to *buildingInstallations*.

All the converted geometries become *lod3MultiSurface*, which is a correct conversion, according to the features of current practice for CityGML data. However, mainly triangulated surfaces are generated and the geometry is not completely controlled, in some cases: each solid is converted into multi-surfaces, which are supposed to represent the same shape (this change is almost successful in the roof slabs representation, although it is possible that the geometry is not completely closed), but in most cases they are converted in a bunch of triangles probably also duplicated (See Figures 6–9).

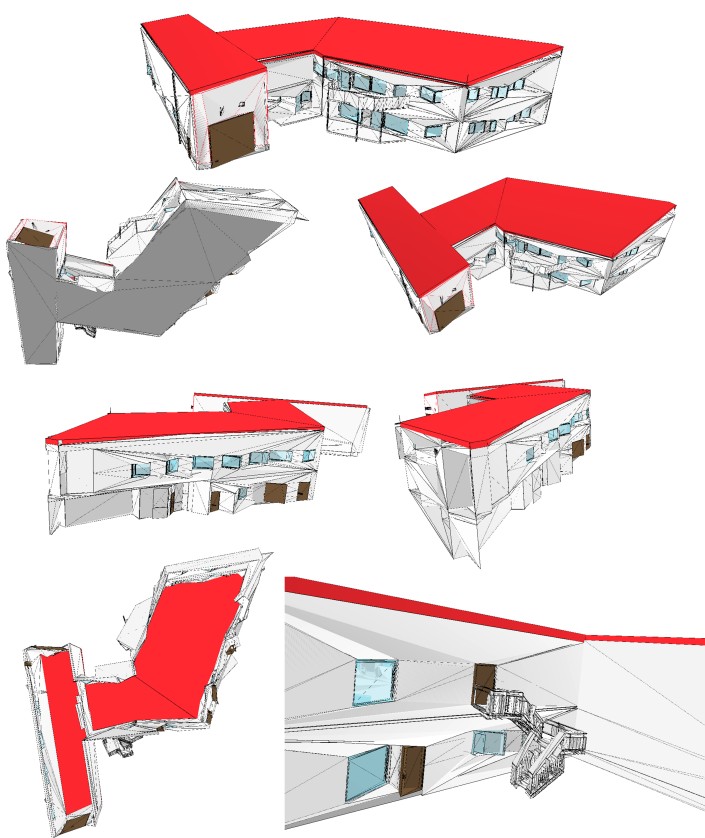

**Figure 6.** Views of the Myran model converted to CityGML by the test AGIS-FME-RVTr-L1, visualized in azul.

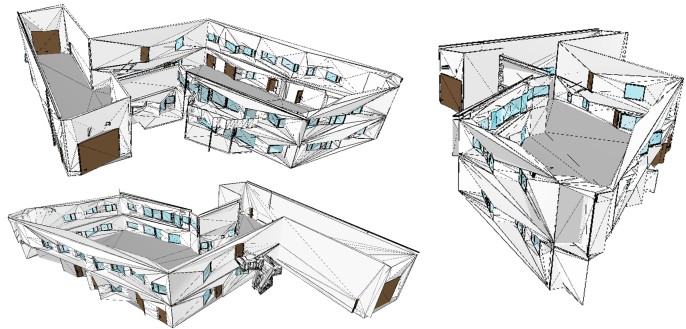

**Figure 7.** Views of the Myran model converted to CityGML by the test AGIS-FME-IFCr-L1, visualized in azul. In this case, the roof is missing.

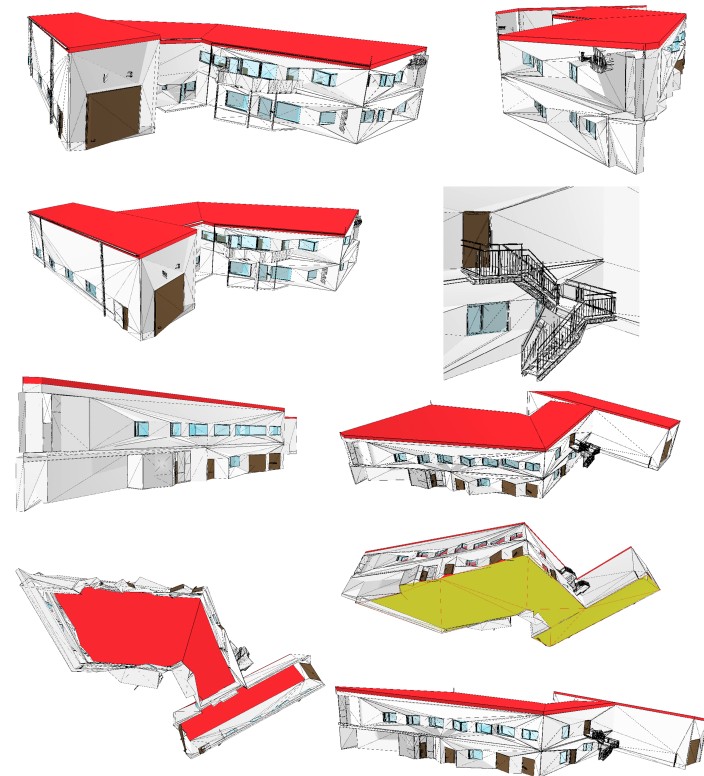

**Figure 8.** Views of the Myran model converted to CityGML by the test FME-RVTr-L1, visualized in azul.

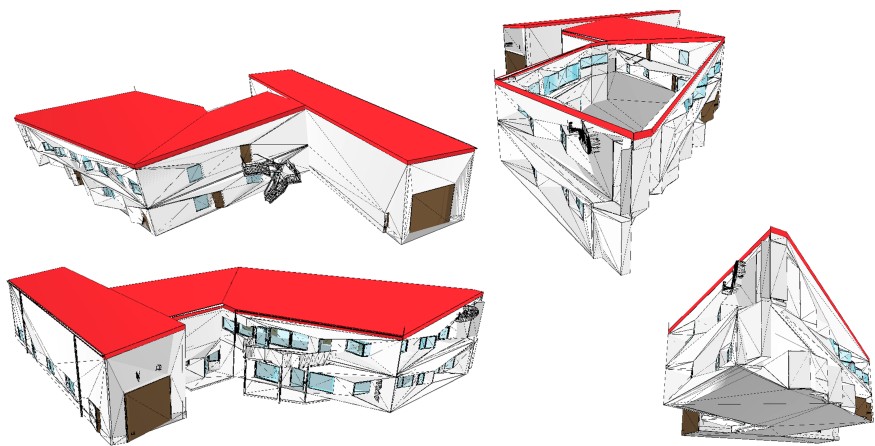

**Figure 9.** Views of the Myran model converted to CityGML by the test FME-IFCr-L1, visualized in azul.

Notwithstanding the residual flaws, these last tests are the closest to achieving a proper conversion result. They used a complex transformation workflow composed by many steps (filtering, selection, mapping, transformations)—see the details and download the built workspaces among the delivered answers by participants at https://www.dropbox.com/s/5oof86wys9db6e7/Task4_DeliveredResults_human.docx.

*IFC2CityGML* is also achieving a nice result, which is compliant to the kind of representation allowed to the version 3 of CityGML, despite adopting a different representation paradigm from the one usually adopted in 3D city models.

*5.2. CityGML to IFC Conversions*

In Table 13, the software tested for the conversions from CityGML to IFC in the Task 4 of the benchmark is summarised. *CityGML2IFC* and *FZK Viewers* were tested with all the three available files, whilst *FME* was tested only with the *BuildingsLoD3.gml* with the *Quick Translator* option and with the *RotterdamLoD12.gml* by means of a more complex workspace.

**Table 13.** Software tested for conversions from CityGML to IFC (Task 4).

|  | Open Source | Proprietary | Freeware |
|---|---|---|---|
| **GIS Software** [1] | | | |
| **'Extended' 3D viewers** [2] | | | FZK Viewer [3] [L1]—2 tests |
| **ETL and conversion software** [4] | CityGML2IFC [5] [L4] | FME Desktop [6]: 1 test using Quick translator [L1]—1 test using a complex workspace [L3] | |
| **BIM software** [7] | | | |

[1] GIS combine different kind of geodata and layers and make analysis on them, structured in a database, in a holistic system. [2] Software originally developed for visualising the 3D semantic models, including georeferencing, and query them. They were (sometimes later) extended with new functions for applying symbology or making simple analysis. [3] https://www.iai.kit.edu/1648.php. [4] Extract, Transform and Load (ETL) software, and conversion software, are able to apply transformations or computations to data. [5] https://github.com/nsalheb/CityGML2IFC. [6] https://www.safe.com. [7] Intended to design buildings or infrastructures according to the the Building Information Modelling methods.

Looking at the results of the conversion, it is possible to notice how generally a change in the format occurs, with little transformation from the typical features of 3D city models to the ones typical of BIM. Of course, in this conversion direction, details should be added to the model, which imply choices that are not straightforward and easily generalizable for any conversion (e.g., thickness of the walls, addition of windows and so on). The need of such transformations should be decided according to specific use cases.

5.2.1. BuildingsLoD3.gml Conversion to IFC

In the conversion from CityGML to IFC of the *BuildingsLoD3.gml* file (Table 14), we can notice that the conversion of semantics follows straightforward rules, as it is reasonable. The geometry as well is converted to the most similar kind of geometry in the converted format, without any further processing. Therefore, the surfaces remain surfaces and the solids remain solids.

**Table 14.** Mapping of the CityGML entities to the IFC entities in the conversion of the BuildingLoD3 model.

| Original Entity | Converted Entity |
|---|---|
| bldg:WallSurface | IfcWall |
| bldg:RoofSurface | IfcSlab[BaseRoof] |
| bldg:Building | IfcBuilding |
| bldg:Door | IfcDoor |
| bldg:Window | IfcWindow |
| bldg:GroundSurface | IfcSlab[BaseSlab] |

Moreover, the roof of a small protruding part at the entrance of one building is represented as *bldg:outerFloorSurface* in the CityGML model, and is converted to *IfcBuildingElementProxy*.

The chimneys are represented by means of *bldg:BuildingInstallation*, which is in turn a composition of *bldg:Wall* and *bldg:ClosureSurface*. They are completely missing in some of the converted models by *FZKViewer* and the one using *FME Quick Translator*.

The mapping was the same for all the software, except for the cited minor changes, and the export to IFC v.4 and v.2x3 give the same results.

The case of *CityGML2IFC* is a bit different, since the tool is developed to manage LoD2 CityGML data. Therefore, in this case doors and windows lose their semantics and their identity as an object, being included in the respective walls (Figure 10). The only entities remaining here are *IfcWalls*, from the conversion of *bldg:Walls* and including also *bldg:Doors*, *bldg:Windows* and *bldg:Installations*; *IfcRoof*, coming from the conversion of *bldg:RoofSurface*, and *IfcSlab* [BaseSlab, GroundSlab] coming from the conversion of the *bldg:GroundSurface*. The *Bldg:OuterFloorSurface* element is lost.

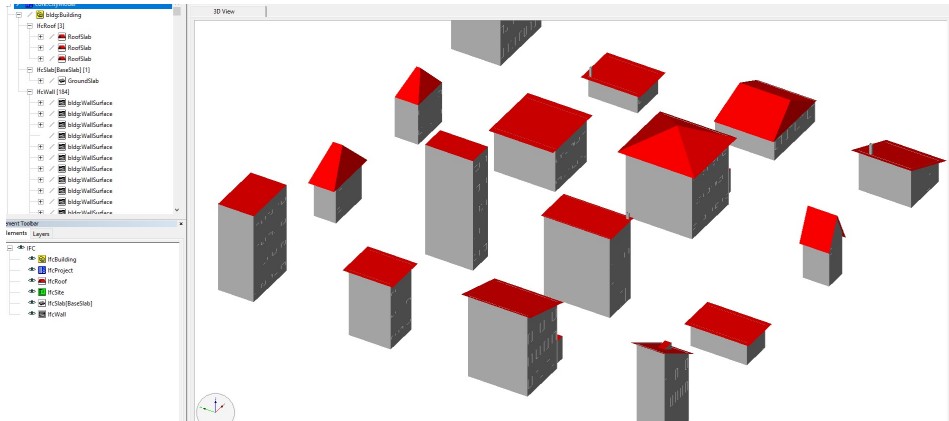

**Figure 10.** BuildingsLoD3.gml model converted to IFC by the *CityGML2IFC* tool, visualized in the *FZKViewer*.

### 5.2.2. RotterdamLoD12 Conversion to IFC

The *RotterdamLoD12.gml* model was converted by the *FZK Viewer*, *FME*, by means of a complex built workspace and *CityGML2IFC* tool.

The Rotterdam models converted by the *FZK Viewer* could only be opened by *FZK Viewer* itself (which gives syntax errors though); it makes other software crash before opening, instead (e.g., *BIMVision* and *RDF IfcViewer*).

After the conversion, many surfaces (both *bldg:walls*, some *bldg:groundSurfaces* and some *bldg:roof parts*) are now *IfcBuildingElementProxy*. Some of them could be the LoD1 representation of the dataset, which was included in the dataset together with the LoD2. The isolation of the geometry having the most suitable LoD gives an additional complexity for this kind of multi-LoD dataset. Probably the LoD2 is the most suitable, since closer to the BIM detail, although it should be finally decided based on use cases.

In the *CityGML2IFC* test, it was already reported in the delivered results that the software totally ignored the LoD1 part of the file. It successfully converts parts of LoD2 data, which is reasonable since being developed to convert from LoD2 CityGML data. However, there was a further problem, probably related to the conversion of the parts of the data where LoD1 and LoD2 overlap. Some parts could therefore be missing, but the remaining objects are consistent: *bldg:wallSurface* to *IfcWall*; *bldg:GroundSurface* to *IfcSlab* and *bldg:RoofSurface* to *IfcRoof*.

In the *FME* workspace test, a processing of the geometry in order to extrude the faces making them solids was applied, as explained in the delivered description. This is a step forward towards building a consistent representation paradigm with BIM. However, such thickness is not always visible in the model, as inspected with RDF IfcViewer (http://rdf.bg/product-list/ifc-engine/ifc-viewer/). The semantic mapping of entities in this case is quite consistent, resulting in *IfcWalls*, *IfcSlabs*, *IfcRoof* and *IfcSite* generated in the area surrounding the buildings. There are some remaining inaccuracies, for example some *IfcSlabs* are used to represent walls (Figure 11a). In addition, the representation of *IfcSpaces* is inconsistent: they are generated only in a part of the buildings and in some cases the

volumes do not fit precisely into the walls but are instead slightly different, intersecting or overlapping them in some cases (Figure 11b).

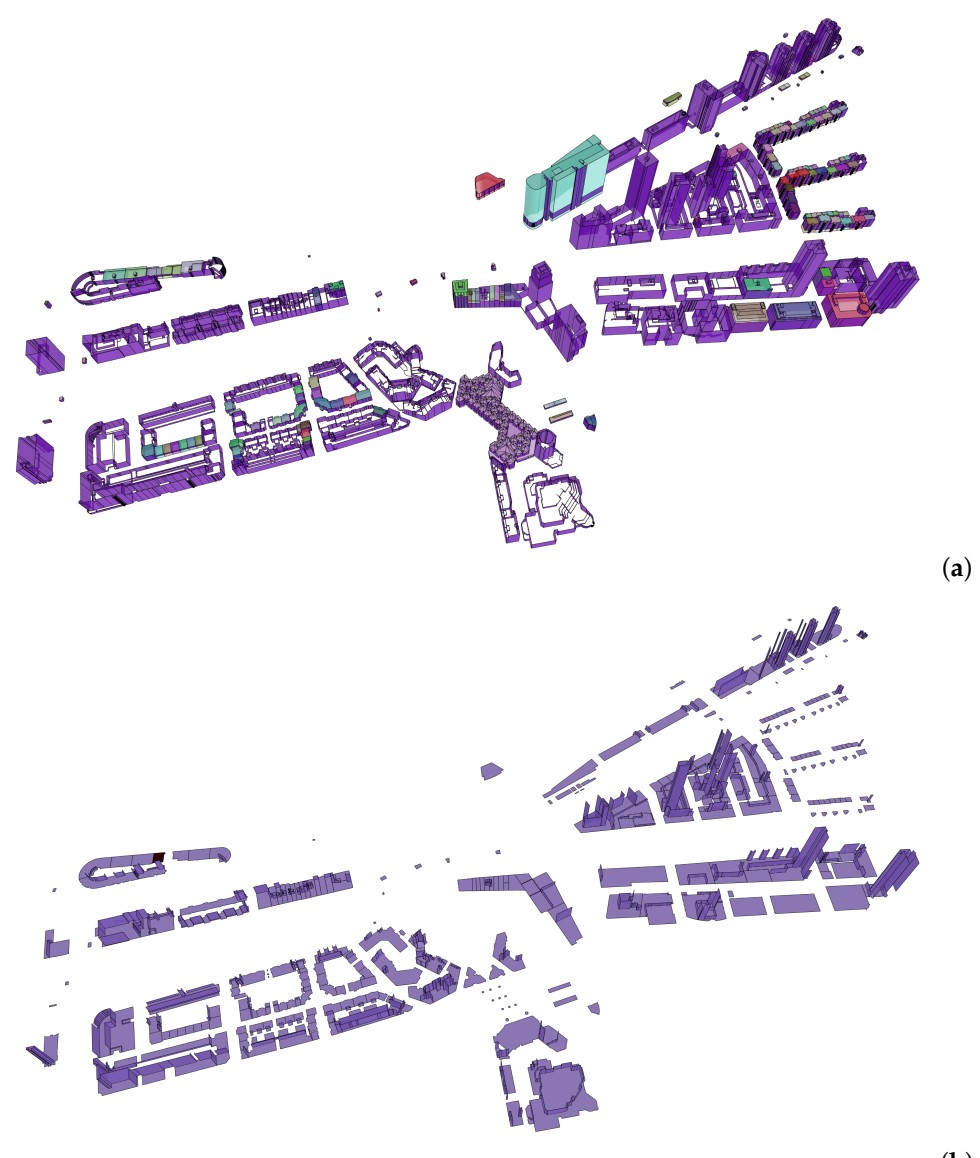

(**a**)

(**b**)

**Figure 11.** Result of FME conversion visualized in RDF IfcViewer. (**a**) IfcWalls in purple and IfcSpaces with various colours; (**b**) IfcSlabs.

The geometry is in all the cases converted from the GML *MultiSurface* to the IFC *SurfaceModel*. However, although the converted *RotterdamLoD12.gml* datasets can be opened in some 3D viewers (with warnings and reported errors sometimes), they cannot be opened for example in *Autodesk Revit*, which is one of the most used software. The result of the conversion made by *FME 2019* can be opened in *Revit*, although only the *IfcSite* is shown, without the *IfcBuildings*. It is of course a big issue if such datasets are supposed to be used as reference in software for design.

5.2.3. Amsterdam Conversion to IFC

The conversion of the *amsterdam.gml* model was tested with the *FZK Viewer*, to both IFC v.2x3 and v.4 and *FME Quick translator*.

All the converted models have quite identical characteristics. They presents the buildings exported as IfcBuildings and the rest of entities as IfcBuildingElementProxy, as fair for the IFC v.2x3, since other

entities explicitly related to infrastructures are not yet in the model. The semantics of the objects remain in the Name attribute of each entity. In addition, the other attributes are converted and associated to the objects correctly.

However, the geometries are almost all flattened to 2D shapes, as visualized in RDF IfcViewer (Figure 12).

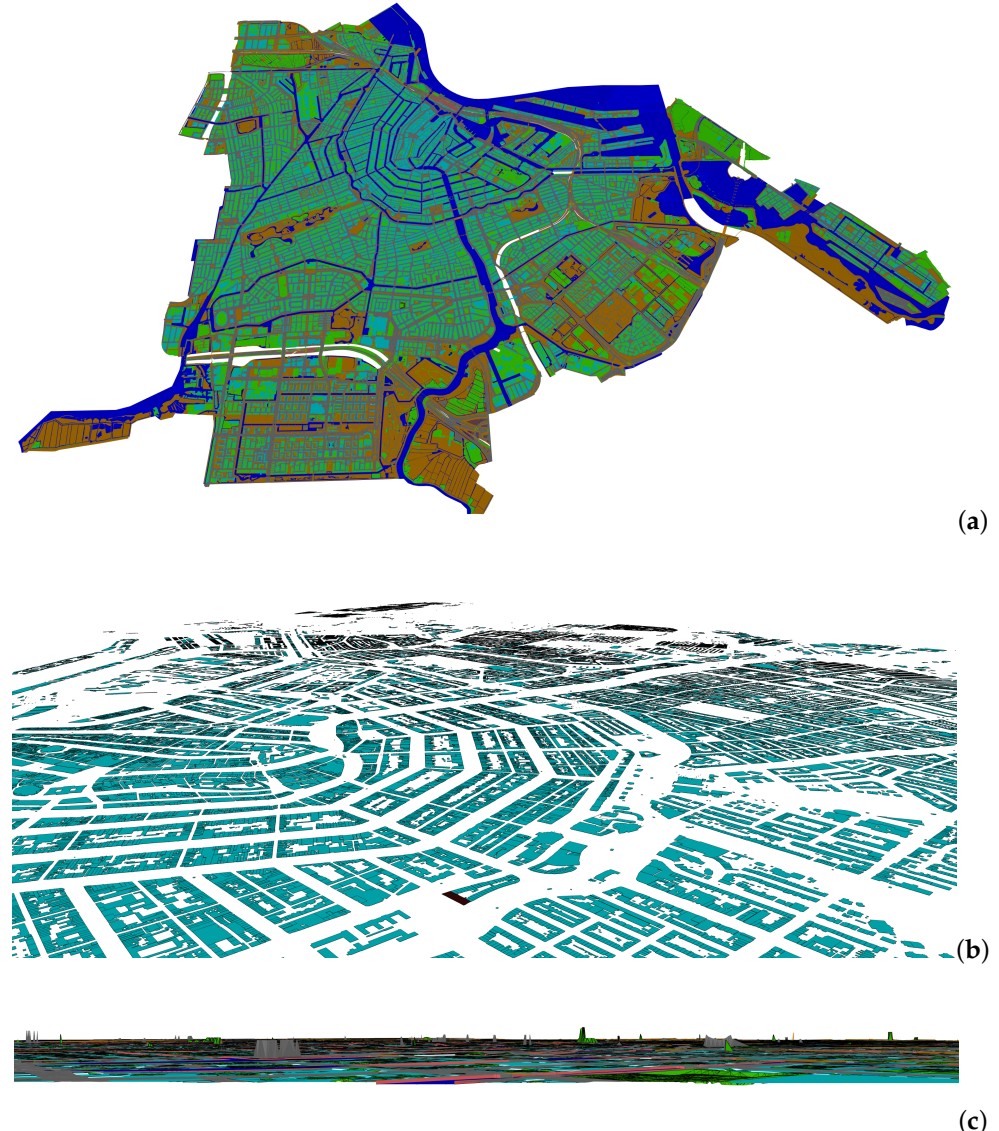

(**a**)

(**b**)

(**c**)

**Figure 12.** Result of FZK conversion of the *amsterdam.gml* model to IFC 2x3, visualized in RDF IfcViewer. (**a**) is the full model, (**b**) the resulting buildings and (**c**) is the side view of the model, where it is possible to appreciate the lack of heights in almost all the objects.

This is probably not a problem, since the use cases requiring the whole city to be used in IFC are not so usual yet, although, for example, infrastructure design could have an advantage from it. In the results, some participants observed that the Amsterdam dataset is large, with many objects. Therefore, if the focus of conversion is only buildings, the other entities should be removed, since they make the conversion process slow and software unresponsive. However, city elements such as the roads, the terrain and even city furniture could be probably among the most relevant object to be considered as reference for buildings and infrastructure design.

*5.3. Conversions Discussion*

The results of both conversions from IFC to CityGML and from CityGML to IFC show how it is quite difficult to perform a complete and consistent conversion from one kind of model to the other one. This considers both the interoperability needs of producing a valid output format and the harmonization issues of making the features of the input model homogeneous with the output one.

One of the closest attempts was performed by *IFC2CityGML*, which has the advantage of the CityGML v.3 representation paradigm allowing the full inclusion of all the BIM concepts within the model without the need to convert them to geo-concepts.

Other good examples are the two ones employing *FME*, as described within the Section 5.1.4, considering the transformation of the solid geometry typical of BIM to the surfaces facing and enclosing the outside part of the building. One of them (FME19-L3) achieved the conversion to CityGML LoD4 with nice results with respect to both geometry and semantics, by developing a very complex processing model.

Others showed how challenging it is to implement a methodology selecting only the necessary parts of the BIM, transform them to a different representation paradigm, by maintaining correctly mapped semantics, and fix it to comply with validity criteria of both geometry and semantics, in association to the application of a generalization to CityGML LoD3.

One of the final questions to participants was about suggestions to improve the input IFC files in order to obtain better conversions. Among those, the participants listed:

- Correct georeferencing;
- a property that states whether an entity has a volumetric or surface geometry;
- modelling interior and exterior separately;
- a property identifying which floors are above ground;
- openings included in wall and slab entities.
- Correct grouping in storeys (e.g., the *Myran.ifc* dataset has correct semantic information w.r.t. attributes but not for the relationships among entities i.e. roof of one storey is part of another storey, a whole storey is defined for just plumbing and beam elements which are not just problematic for analysis but also semantically incorrect);
- avoid geometries overlap;
- correct and consistent use of *IfcSpaces*.

The conversion CityGML to IFC counted on a limited experience with respect to the conversion IFC to CityGML, because of the fact that most of initial efforts were dedicated to the use of BIM information to be integrated in 3D city models, and not the other way around, which is also implying more formal challenges (not just the technical ones).

In addition, it is important to note that in general, IFC to CityGML is a lossy conversion. IFC schema and models typically have more fine grained details than city scale CityGML models do. Therefore, it is reasonable to expect better results from an IFC (BIM) to CityGML (city model) conversion than the other way around (CityGML to IFC).

The output is generally generated by tools without a deeper consideration of complex BIM and IFC norms or best practices. The output data should be compliant with the minimal requirements of the IFC standard, but is lacking in terms of the logical structure and relationships that a typical domain expert would know.

Most of conversions from CityGML to IFC produce boundary representation models which are probably valid within the IFC model, although different parameters should be taken into account in this regard. For example, what use case will they be used for? What kind of information does this use case need? Within it, a representation of extremely general *IfcRoofs* and *IfcWalls* are useful? Would the representation of windows help or not? What kind of geometry would be necessary? And so on.

The generation of a BIM with its own proper representation paradigm (walls with thickness and maybe materials, slabs and so on) would need a way more complex methodology, adopting a combination of more sophisticated technologies and studies supporting them.

As some of the participants discussed, additional support could come from the storage of relevant to typical BIM requirements in the CityGML dataset, such as materials, wall thickness or other classification information.

Within the benchmark, not the whole set of proposed procedures in literature could be tested. This would be an extremely difficult task since many procedures are only described theoretically or do not make their implementations publicly available. However, the measures for potentially reducing this gap were taken by inviting everyone to join in the tests (which would allow developers to test their tools without making them freely available). The tested procedures are therefore the ones most available at the moment for any user interested in making conversions, which was the actual scope of interest of the benchmark.

It is also worth noting that very few open source tools have been, which largely reflects the lack of open source tools currently available for this task. While this weakens the benchmark results, this is more of a shortcoming of the state of the art than a shortcoming of the benchmark itself.

The study could enlighten, first of all, that some geometries encounter difficulties in being converted, as it is apparent from the tests with the *IFCgeometries.ifc* dataset (Section 5.1.1). Moreover, it made the need clear of an overall methodology considering not only the interoperability issue but also the harmonization side of the problem, which is not completely tackled yet.

In addition, the multi-LoD files present more challenges, since the geometries from different LoD can overlap, notwithstanding being part of consistent and valid models. Therefore an initial step selecting only the LoD to be considered for the conversion should help, in most of cases.

It is clear how an aware composition of intermediate processes (such as FME transformers or self-developed algorithms) is necessary to reach high quality results.

The steps involved in the most successful conversions are generally summarized as: feature mapping—attributes mapping—geometry mapping and processing—mapping of relationships and properties.

That implies that a sufficient knowledge of both standards (IFC and CityGML) and of the tool is necessary.

## 6. Conclusions

Within the GeoBIM benchmark, the state of implementation of the CityGML and IFC standards was investigated and tested. In this paper, the part of the study more directly concerning the integration of 3D city models with BIM was described, including the test of the tools allowing the georeferencing of IFC and conversion procedures in both directions from IFC to CityGML and from CityGML to IFC.

The results of the Task 2 of the benchmark (Section 4) showed what tools are available to apply georeferencing to IFC data. The major flaw is found as the lack of control on the way georeferencing is stored within the IFC file. A collaboration among use cases stakeholders and researchers, developers and standardisation organizations should be necessary to decide on agreements in that respect. Furthermore, the delivered results did not use the available IFC v.4 entities to export the IFC georeferenced files in v.4.

The conversions task (Task 4) of the benchmark outlined similar flaws. On the one hand little explicit constraints are stated by the standards, which makes the validity and assessing criteria of the resulting models more based on the current practice than on the standards themselves. Moreover, some use cases-based consideration and parameters should be defined in order to outline clearly the transformations needed. On the other hand, the more successful conversion procedures tended to consider and model a complex architecture where mapping and transformations of semantics and geometric features of the data are taken into account. The improvement of them by considering needs from practice and use cases is a topic that needs further attention.

An additional push would be the definition and control over validity criteria for input models. Similarly, another area that revealed gaps in the standards was clear methods for validating output model with respect to the input model. In the context of CityGML, at least the output GML can

be validated against the CityGML application schemas. However, for IFC, no such such automatic validation facility exists. In fact, a separate study could be conducted by focusing on the whole area of validation rules and methods which itself deserves in depth exploration.

The limitations of the study, as well as its strength, lie in the involvement of voluntary participants to make the tests. This potentially opened the participation to anyone developing a suitable procedure for one of the tasks, with an inclusive approach. However, it required them to actively join and invest some time in it, which could have hindered thorough participation. Contrary to the measures adopted for Task 1 and Task 3, for which the tests were integrated about the still uncovered tools, in these cases the issues were too complex, and with the most spread off-the-shelf tools already considered, it was not judged essential to try other codes potentially available in literature (besides probably not straightforward).

While the participants' tests are not sufficient to be considered as authoritative proof at this point, we hope that they can be treated as data points to show that much work remains to be done regarding the development of georeferencing and conversion tools.

This study identifies specifically what are the areas of the issues to be developed further, both in software tools and through improved standards, to effectively support the integration: higher and better awareness and control of the georeferencing method used to be transparently implemented in the tools; definition of validity criteria and constraints for the produced models both for 3D city models and BIMs, based on use-cases tailored parameters. Especially this last need could be useful both in the case of conversions and in the bare modelling itself. One definitely more complex step lies in the development of a comprehensive methodology considering such parameters to actually transform one kind of model into the other one, with both format and kind of features. The investigated issues, georeferencing of IFC and conversions, are mainly due to the misalignment of the formal background representing the concepts, requirements and criteria to be respected and the implementation of them in tools.

Regarding software, we would encourage tool developers to work on implementing the LoGeoRef levels in their georeferencing tools. Similarly, developers working on conversion procedures should take into account the particularities on the two standards (both formal and unwritten). While doing a 1:1 mapping of the geometry is useful in certain situations, it is also insufficient in many situations, which requires geometry and semantic processing to fit with the other standard.

These points, and especially the need of coordination between research, standardisation efforts and implementations, will be addressed in future research towards the integration of geoinformation and 3D city models with BIM.

**Author Contributions:** Conceptualization, Francesca Noardo, Lars Harrie, Ken Arroyo Ohori and Jantien Stoter; data curation, Francesca Noardo and Thomas Krijnen; formal analysis, Francesca Noardo; funding acquisition, Francesca Noardo, Lars Harrie, Ken Arroyo Ohori, Filip Biljecki, Claire Ellul, and Jantien Stoter; investigation, Francesca Noardo, Lars Harrie, Ken Arroyo Ohori, Filip Biljecki, Claire Ellul, Thomas Krijnen, Helen Eriksson, Dogus Guler, Dean Hintz, Mojgan A. Jadidi, Maria Pla, Santi Sanchez, Ville-Pekka Soini, Rudi Stouffs, Jernej Tekavec and Jantien Stoter; methodology, Francesca Noardo, Lars Harrie, Ken Arroyo Ohori, Filip Biljecki, Claire Ellul and Jantien Stoter; project administration, Francesca Noardo; writing—original draft, Francesca Noardo; writing—review and editing, Lars Harrie, Ken Arroyo Ohori, Filip Biljecki, Claire Ellul, Thomas Krijnen, Helen Eriksson, Dogus Guler, Dean Hintz, Mojgan A. Jadidi, Maria Pla, Santi Sanchez, Ville-Pekka Soini, Rudi Stouffs, Jernej Tekavec and Jantien Stoter. All authors have read and agreed to the published version of the manuscript.

**Funding:** This research was funded by the International Society for Photogrammetry and Remote Sensing (ISPRS)—Scientific Initiatives 2019 and the European Association for Spatial Data Research (EuroSDR). This project has also received funding from the European Research Council (ERC) under the European Union's Horizon 2020 Research & Innovation Programme (grant agreement no. 677312, Urban modelling in higher dimensions) and from European Union's Horizon 2020 Research & Innovation Programme Marie Skłodowska-Curie (grant agreement No. 707404, Multisource Spatial data Integration for smart City Applications).

**Acknowledgments:** This work was possible thanks to the collaboration of the whole GeoBIM benchmark team (with their work as in-kind contribution to the project), all the data providers, the participants making the tests, listed in the GeoBIM benchmark website (https://3d.bk.tudelft.nl/projects/geobim-benchmark/participants.html). The IFC geometries dataset was specially prepared for this project by Thomas Krijnen.

**Conflicts of Interest:** The authors declare no conflict of interest. The funders had no role in the design of the study; in the collection, analyses, or interpretation of data; in the writing of the manuscript, or in the decision to publish the results.

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
