# Peer review of "Tools for BIM-GIS Integration (IFC Georeferencing and Conversions): Results from the GeoBIM Benchmark 2019"

_ijgi, doi:10.3390/ijgi9090502_

Round 1
Reviewer 1 Report
This manuscript discusses a benchmark framework for evaluating the performance of BIM-GIS integration process that includes the IFC georeferencing and conversions. This topic is quite interesting and meaningful for the 3D urban model harmonization. However, there are still flaws in the current manuscript: 1. The comparing data conversion tools are lack of many open source candidates, and sometimes both three categories of software are lack of candidates. This situation will reduce the authorities of this benchmarking framework. Besides, due to the black box implementation mechanism of FME and ArcGIS, their conversion performance differences could not be analyzed in-depth. 2. The volume of the test dataset is also limited, considering the large scale of potential covering topics. Several typical scenes are not enough for thoroughly providing every aspect of the metric of comparing toolsets for converting data between BIM and GIS. 3. As described in the manuscript, the inconsistently designed purpose of IFC model and CityGML model fundamentally determines that there are data objects not able to be converted without any information losses. Therefore, if the data accuracy is crucial, then reserving original BIM models is a better choice than using the GIS models and this accuracy loss GIS models could be generated to provide a general glance of the landscape. Thus, is it possible to develop a specific part of the benchmark to evaluate the data loss percentages under this circumstance?
Author Response
This manuscript discusses a benchmark framework for evaluating the performance of BIM-GIS integration process that includes the IFC georeferencing and conversions. This topic is quite interesting and meaningful for the 3D urban model harmonization. However, there are still flaws in the current manuscript:
> Thank you for the comments
1. The comparing data conversion tools are lack of many open source candidates, and sometimes both three categories of software are lack of candidates. This situation will reduce the authorities of this benchmarking framework. Besides, due to the black box implementation mechanism of FME and ArcGIS, their conversion performance differences could not be analyzed in-depth.
> The software categories were defined for the whole benchmark. For the other benchmark tasks, there are several open source alternatives (eg BimServer and FreeCAD for Task 1/BIM, or QGIS and 3DCityDB for Task 3/Geo). Unfortunately there is a general lack of open source tools that are able to perform conversions, which results in these empty categories.
> While this is beyond the scope of the benchmark to fix, we have added a paragraph in the discussion that explains this issue.
2. The volume of the test dataset is also limited, considering the large scale of potential covering topics. Several typical scenes are not enough for thoroughly providing every aspect of the metric of comparing toolsets for converting data between BIM and GIS.
> This is certainly true, but even if the results cannot be fully authoritative currently, we believe that it is best to have the benchmark results available and to treat it as a set of data points. Further research by other authors/participants can tackle the remaining gaps and strengthen the results over time.
> It is also worth pointing out that due to the quick changes in software capabilities and versions, we believe that waiting for further results before attempting the publication is not desirable.
3. As described in the manuscript, the inconsistently designed purpose of IFC model and CityGML model fundamentally determines that there are data objects not able to be converted without any information losses. Therefore, if the data accuracy is crucial, then reserving original BIM models is a better choice than using the GIS models and this accuracy loss GIS models could be generated to provide a general glance of the landscape. Thus, is it possible to develop a specific part of the benchmark to evaluate the data loss percentages under this circumstance?
> Considering that the conversion process in software is usually done by first loading IFC/CityGML into the software's internal data model, from which it is then exported to the other format, the data loss can be measured more accurately through the import and export to the same format. This is done in Tasks 1 and 3 of the benchmark.
> Measuring the data loss across formats is more difficult, and in our experience really depends on the application. For some applications the loss in semantics is not really important (eg giving visual context to a BIM model through a 3C city model), whereas more specialised applications will need the preservation of specific classes.
> It is also worth saying that it is difficult to obtain harder numbers from this type of benchmark. Since it was (by design) an open experiment with users of different experience on different hardware and settings, we refrained from asking about very specific numbers.
Reviewer 2 Report
This is a valuable research to investigate the tools for BIM-GIS integration. The authors did in-depth evaluations and comparisons of different tools for different tasks. The reviewer has some minor questions:
- What is the background and experience for the participants in this study?
- What would be the suggested directions for future development for these tools?
Author Response
This is a valuable research to investigate the tools for BIM-GIS integration. The authors did in-depth evaluations and comparisons of different tools for different tasks. The reviewer has some minor questions:
> Thank you for the comments
What is the background and experience for the participants in this study?
> Comprehensive background on the participants was not systematically collected within the benchmark, but they are a mix of students, practitioners, software developers and researchers. The specific experience of the participants regarding the tools is available in Table 9 (L1-L4).
What would be the suggested directions for future development for these tools?
> Regarding georeferencing tools, the biggest issue is the lack of standard methods that are supported everywhere and are accurate. Such methods should ideally be implemented in all software. The starting point for these should probably be the LoGeoRef levels.
> Regarding conversion tools, the issues are many, but we would say that the misalignement between the two standards' concepts, requirements and criteria are at the root. These should probably be tackled through improvements in the standards first, since this hinders tool development. However, developing tools that are aware of the conventions of the two standards (as opposed to a 1:1 mapping of the geometry with generic semantics) is a good first step.
> We have added a paragraph in the conclusions with recommendations for tool developers.
Reviewer 3 Report
General Comments
This paper presents a comprehensive review of critical points regarding harmonization/interface of components across BIM and geographical GIS environment, with detailed practical benchmark from specialists and end users.
The paper is very well written and presented and in this author’s opinion deserves publication.
The only weak point of the approach is the scientific reliability of the survey, as it is mentioned in the disclaimer in the conclusions. Perhaps the authors could clarify in a more scientifically quantitative way the statistical significance (if any) of the qualitative end users feedback.
Other Comments
Abbreviations are somehow tricky to follow, particularly at the beginning of the maniuscript. For example City GML, although established and well known, it is introduced later on, and should be define before.
In general the scientific writing of the paper can be improved, smoothening the discussion of technical benchmark points (tasks, etc), as in several points it prevails more as a project report.
Author Response
General Comments
This paper presents a comprehensive review of critical points regarding harmonization/interface of components across BIM and geographical GIS environment, with detailed practical benchmark from specialists and end users.
The paper is very well written and presented and in this author’s opinion deserves publication.
> Thank you for the comments
The only weak point of the approach is the scientific reliability of the survey, as it is mentioned in the disclaimer in the conclusions. Perhaps the authors could clarify in a more scientifically quantitative way the statistical significance (if any) of the qualitative end users feedback.
> It is difficult to provide a quantitative result at this point, both due to the number of participants and the open-ended nature of their results. However, we have added some points in the discussion and harder conclusions where we are certain.
> Mainly, that it seems clear that georeferencing and conversion tools are not adequate at this point. More work is clearly needed on this front.
Other Comments
Abbreviations are somehow tricky to follow, particularly at the beginning of the maniuscript. For example City GML, although established and well known, it is introduced later on, and should be define before.
> We have gone through the paper and added the remaining acronyms' full names.
In general the scientific writing of the paper can be improved, smoothening the discussion of technical benchmark points (tasks, etc), as in several points it prevails more as a project report.
> We have gone through the text and made small improvementes throughout.